# Optimal Transport Barycenter via Nonconvex Concave Minimax Optimization

## Abstract

The optimal transport barycenter (a.k.a. Wasserstein barycenter) is a fundamental notion of averaging that extends from the Euclidean space to the Wasserstein space of probability distributions. Computation of the *unregularized* barycenter for discretized probability distributions on point clouds is a challenging task when the domain dimension $d > 1$. Most practical algorithms for approximating the barycenter problem are based on entropic regularization. In this paper, we introduce a nearly linear time $O(m \log m)$ and linear space complexity $O(m)$ primal-dual algorithm, the *Wasserstein-Descent* $\dot{\mathbb{H}}^1$-*Ascent* (WDHA) algorithm, for computing the *exact* barycenter when the input probability density functions are discretized on an $m$-point grid. The key success of the WDHA algorithm hinges on alternating between two different yet closely related Wasserstein and Sobolev optimization geometries for the primal barycenter and dual Kantorovich potential subproblems. Under reasonable assumptions, we establish the convergence rate and iteration complexity of WDHA to its stationary point when the step size is appropriately chosen. Superior computational efficacy, scalability, and accuracy over the existing Sinkhorn-type algorithms are demonstrated on high-resolution (e.g., $1024 \times 1024$ images) 2D synthetic and real data.

## 1 Introduction

The Wasserstein barycenter, introduced by Agueh & Carlier (2011) based on the theory of optimal transport (OT), extends the notion of a Euclidean average to measure-valued data, thus representing the "average" of a set of probability measures. Direct applications of the Wasserstein barycenter include smooth interpolation between shapes (Solomon et al., 2015), texture mixing (Rabin et al., 2012), and averaging of neuroimaging data (Gramfort et al., 2015), among others. More importantly, the computation of the Wasserstein barycenter often serves as a key stepping stone to derive more advanced machine learning and statistical algorithms. For instance, centroid-based methods for clustering distributions rely on the computation of Wasserstein barycenters (Cuturi & Doucet, 2014; Zhuang et al., 2022). Additionally, regression models and statistical inference methods for distributional data that utilize Wasserstein geometry often employ the barycenter as an "anchor" measure to map distributions to a linear tangent space (Dubey & Müller, 2020; Zhang et al., 2022; Chen et al., 2023; Zhu & Müller, 2023; Zhu & Müller, 2024; Jiang et al., 2024).

Despite the widespread applications, computationally efficient or even scalable algorithms of the Wasserstein barycenter with theoretical guarantees remain to be developed. Existing approaches to compute the Wasserstein barycenter of a collection of probability density functions $\mu_1, \ldots, \mu_n$ in $\mathbb{R}^d$ rely on a Wasserstein analog of the gradient descent algorithm, which requires to compute $n$ OT maps per iteration (Zemel & Panaretos, 2019; Chewi et al., 2020). Álvarez-Esteban et al. (2016) propose a fixed point approach that is effective for any location-scatter family. Classical OT solvers, such as Hungarian method (Kuhn, 1955), auction algorithm (Bertsekas & Castanon, 1989) and transportation simplex (Luenberger & Ye, 2008), scale poorly for even moderately mesh-sized problems. This presents a substantial computational barrier for computing the barycenter of multivariate distributions. On the other hand, various regularized barycenters have been proposed to mitigate the computational difficulty (Li et al., 2020a; Janati et al., 2020; Bigot et al., 2019; Carlier et al., 2021; Chizat, 2023), and Sinkhorn's algorithm is perhaps one of the most widely used algorithms to compute the entropy-regularized barycenter (Peyré & Cuturi, 2019; Lin et al.,

2022; Carlier, 2022). Also, Li et al. (2020b) propose a new dual formulation for the regularized Wasserstein barycenter problem such that discretizing the support is not needed. However, the computation thrifty of these methods is subject to the approximation accuracy trade-off (Nenna & Pegon, 2024). In low-dimensional settings such as 2D images, this is often manifested in visually undesirable blurring effects on the barycentric images. To break the curse of dimensionality, scalable algorithms using input convex neural networks (Korotin et al., 2021) and generative models (Korotin et al., 2022) have been investigated for the Wasserstein barycenter problem.

In this paper, we recast the **unregularized** optimal transport (a.k.a. Wasserstein) barycenter problem as a nonconvex-concave minimax optimization problem and propose a coordinate gradient algorithm, termed as the *Wasserstein-Descent $\dot{\mathbb{H}}^1$-Ascent* (WDHA), which alternates between the Wasserstein and Sobolev spaces. The key innovation of our WDHA algorithm is to combine *two different yet closely related* primal-dual optimization geometries between the primal subproblem for updating barycenter with the Wasserstein gradient and the Kantorovich dual formulation of the Wasserstein distance for updating the potential functions with a homogeneous $\dot{\mathbb{H}}^1$ gradient (cf. Definition in Section 2.2). In contrast with the usual $L^2$ gradient $\nabla$, our choice of the $\dot{\mathbb{H}}^1$ gradient can be interpreted as an isometric dual embedding of the potential function $\phi$ corresponding to the earthmoving effort for pushing a source distribution $\mu$ to a target distribution $\nu$ via $\|\mu - \nu\|^2_{\dot{\mathbb{H}}^{-1}} = \int \|\nabla\phi\|^2_2$. This OT perspective is critical to ensure stability for our $\dot{\mathbb{H}}^1$-gradient ascent subproblem and therefore the convergence of the overall WDHA algorithm.

## 1.1 CONTRIBUTIONS

Current work is among the first to combine the Wasserstein and homogeneous Sobolev gradients to derive a simple, scalable, and accurate primal-dual coordinate gradient algorithm for computing the exact (i.e., unregularized) OT barycenter. The proposed WDHA algorithm is particularly suitable for computing the barycenter for discretized probability density functions on a large $m$-point grid such as images of $1024 \times 1024$ resolution. The proposed WDHA algorithm enjoys strong theoretical properties and empirical performance. The following summarizes our main contributions.

- Discretizing the input probability density functions $\mu_1, \ldots, \mu_n$ onto a grid of $m$ points, the per iteration runtime complexity of our algorithm is $O(m \log m)$ for updating each Kantorovich potential. This is in sharp contrast with the time cost $O(m^3)$ for computing an OT map between two distributions supported on the same grid via linear programming (LP). In addition, the space complexity for the $\dot{\mathbb{H}}^1$ gradient is $O(m)$, which also substantially reduces the LP space complexity $O(m^2)$.
- Under reasonable assumptions, we provide an explicit algorithmic convergence rate and iteration complexity of the WDHA algorithm to its stationary point with appropriately chosen step sizes for the gradient updates. In particular, WDHA achieves the same convergence rate $O(1/T)$ as in the Euclidean nonconvex-concave optimization problems.
- We demonstrate superior numeric accuracy and computational efficacy over Sinkhorn-type algorithms on high-resolution 2D synthetic and real image data, where the standard Wasserstein gradient descent algorithm cannot be practically implemented on such problem size.

For limitations, the current approach is mainly limited to computing the Wasserstein barycenter of 2D or 3D distributions supported on a compact domain.

## 1.2 NOTATIONS

We use $\mathbb{R}^d$, $\mathbb{H}$, and $\mathcal{P}^r_2(\Omega)$ to represent the $d$-dimensional Euclidean space, a Hilbert space, and the Wasserstein space on $\Omega$ respectively. Let $\|\cdot\|_2$ and $\langle\cdot,\cdot\rangle$ ($\|\cdot\|_{\mathbb{H}}$ and $\langle\cdot,\cdot\rangle_{\mathbb{H}}$) denote the Euclidean (Hilbert) norm and inner-product. Given a function $\varphi : \mathbb{R}^d \to \mathbb{R}$ and functionals $\mathcal{I} : \mathbb{H} \to \mathbb{R}$, $\mathcal{F} : \mathcal{P}^r_2 \to \mathbb{R}$, we use $\nabla\varphi$, $\boldsymbol{\nabla}\mathcal{I}$, and $\mathbb{W}\mathcal{F}$ to represent the standard gradient on $\mathbb{R}^d$, the $\mathbb{H}$-gradient, and the Wasserstein gradient respectively. Let $\varphi^* := \sup_y \langle\cdot, y\rangle - \varphi(y)$ denote the convex conjugate of $\varphi$, and $\varphi^{**}$ denote the second convex conjugate. We use id to represent the identity map and the notation $[T] = \{1, 2, \ldots, T\}$. Given two probability measures $\nu$ and $\mu$, let $T^\mu_\nu$ denote the optimal transport map that pushes $\nu$ to $\mu$, and let $\varphi^\mu_\nu$ be the corresponding Kantorovich potential. A more detailed list of notations is provided in the appendix.

## 2 PRELIMINARY

### 2.1 MONGE AND KANTOROVICH OPTIMAL TRANSPORT PROBLEMS

Let $\mathcal{P}_2(\Omega)$ be the set of probability measures on a convex compact set $\Omega \subseteq \mathbb{R}^d$ with finite second-order moments, i.e., it holds that $\int_\Omega \|x\|_2^2 \, \mathrm{d}\mu(x) < \infty$ for any $\mu \in \mathcal{P}_2(\Omega)$. For $\nu, \mu \in \mathcal{P}_2(\Omega)$, the Monge's optimal transport (OT) problem for the quadratic cost can be written as

$$\mathcal{W}_2^2(\nu, \mu) := \inf_{T:T_\#\nu=\mu} \mathcal{M}(T) \quad \text{with} \quad \mathcal{M}(T) := \int_\Omega \frac{1}{2}\|T(x) - x\|_2^2 \, \mathrm{d}\nu(x),$$

where $T_\#\nu$ is the push-forward measure of $\nu$ by $T$, and $\mathcal{W}_2(\nu, \mu)$ is called the 2-Wasserstein distance between $\nu$ and $\mu$. Though the solution of Monge's problem may not exist, its relaxation, the Kantorovich formulation of the optimal transport problem shown below, always admits a solution,

$$\min_{\lambda \in \Pi(\nu,\mu)} \mathcal{K}(\lambda) := \int_{\Omega \times \Omega} \frac{1}{2}\|x - y\|_2^2 \, \mathrm{d}\lambda(x, y),$$

where $\Pi(\nu, \mu)$ is the set of probability measures on $\Omega \times \Omega$ with marginal distributions $\nu$ and $\mu$. The optimal solution $\lambda$ is called the optimal transport plan. When $\nu \in \mathcal{P}_2^r(\Omega)$, the subset of $\mathcal{P}_2(\Omega)$ consisting of all absolutely continuous probability measures (with respect to the Lebesgue measure on $\Omega$), it is known that the solution $T_\nu^\mu$ of Monge's problem exists, and the optimal transport plan is $\lambda = (\mathrm{id}, T_\nu^\mu)_\#\nu$. In this work, we will utilize the following dual form of the Kantorovich's problem,

$$\min_{\lambda \in \Pi(\nu,\mu)} \mathcal{K}(\lambda) = \max_{\varphi:\Omega \to \mathbb{R} \text{ is convex}} \mathcal{I}_\nu^\mu(\varphi)$$

$$\text{with} \quad \mathcal{I}_\nu^\mu(\varphi) := \int_\Omega \frac{\|x\|_2^2}{2} - \varphi(x) \, \mathrm{d}\nu(x) + \int_\Omega \frac{\|x\|_2^2}{2} - \varphi^*(x) \, \mathrm{d}\mu(x), \tag{1}$$

where $\varphi^* : \Omega \to \mathbb{R}$ is the convex conjugate of $\varphi$. Maximizers of the above Kantorovich dual problem are referred to as Kantorovich potentials. Brenier's Theorem states that the Kantorovich potential $\varphi$ is unique when $\nu \in \mathcal{P}_2^r(\Omega)$, and the optimal transport map satisfies $T_\nu^\mu = \mathrm{id} - \nabla\varphi$. More details of optimal transport theory are referred to the monograph (Santambrogio, 2015).

### 2.2 $\dot{\mathbb{H}}^1$ GRADIENT

In this subsection, we review the concept of $\mathbb{H}$ gradient and introduce a $\dot{\mathbb{H}}^1$-gradient ascent approach for finding the maximizers of $\mathcal{I}_\nu^\mu(\varphi)$ proposed by Jacobs & Léger (2020). Gâteaux derivative generalizes the standard notion of a directional derivative to functionals. Given a functional $\mathcal{F} : \mathbb{H} \to \mathbb{R}$ defined on a Hilbert space $\mathbb{H}$ with inner product $\langle \cdot, \cdot \rangle_\mathbb{H}$, the Gâteaux derivative of $\mathcal{F}$ at $\phi \in \mathbb{H}$ in the direction $h \in \mathbb{H}$, denoted by $\delta\mathcal{F}_\phi(h)$, is defined as

$$\delta\mathcal{F}_\phi(h) = \frac{\mathrm{d}}{\mathrm{d}\epsilon}\mathcal{F}(\phi + \epsilon h)\Big|_{\epsilon=0}.$$

Furthermore, the map $\boldsymbol{\nabla}\mathcal{F} : \mathbb{H} \to \mathbb{H}$ is referred to as the $\mathbb{H}$ gradient of $\mathcal{F}$ if $\langle \boldsymbol{\nabla}\mathcal{F}(\phi), h \rangle_\mathbb{H} = \delta\mathcal{F}_\phi(h)$ holds for all $\phi, h \in \mathbb{H}$. When $\mathbb{H}$ is $\mathbb{R}^d$, $\boldsymbol{\nabla}\mathcal{F}$ simplifies to the standard gradient of a function. Now, we consider the following homogeneous Sobolev space,

$$\dot{\mathbb{H}}^1 := \left\{ \varphi : \Omega \to \mathbb{R} \ \middle| \ \int_\Omega \varphi(x) \, \mathrm{d}x = \int_\Omega \frac{\|x\|_2^2}{2} \, \mathrm{d}x, \int_\Omega \|\nabla\varphi(x)\|_2^2 \, \mathrm{d}x < \infty \right\},$$

where $\nabla\varphi$ is the weak derivative of $\varphi$ (Evans, 2022). It is shown that $\dot{\mathbb{H}}^1$ is a Hilbert space with the inner product $\langle \varphi_1, \varphi_2 \rangle_{\dot{\mathbb{H}}^1} = \int_\Omega \langle \nabla\varphi_1(x), \nabla\varphi_2(x) \rangle \, \mathrm{d}x$. As demonstrated by Jacobs & Léger (2020), the $\dot{\mathbb{H}}^1$ gradient of $\mathcal{I}_\nu^\mu : \dot{\mathbb{H}}^1 \to \mathbb{R}$ is given by

$$\boldsymbol{\nabla}\mathcal{I}_\nu^\mu(\varphi) = (-\Delta)^{-1}(-\nu + (\nabla\varphi^*)_\#\mu), \tag{2}$$

where $(-\Delta)^{-1}$ denotes the negative inverse Laplacian operator with zero Neumann boundary conditions. We can always assume $\boldsymbol{\nabla}\mathcal{I}_\nu^\mu(\varphi) \in \dot{\mathbb{H}}^1$ by noting that adding a constant to a function does not affect its Laplacian. Note that $\mathcal{I}_\nu^\mu : \dot{\mathbb{H}}^1 \to \mathbb{R}$ is a concave functional. With the definition

of $\dot{\mathbb{H}}^1$ gradient, the following $\dot{\mathbb{H}}^1$-gradient ascent algorithm (Algorithm 1) can be applied to solve $\max_\varphi \mathcal{I}_\nu^\mu(\varphi)$, where $\varphi^*$ represents the convex conjugate of $\varphi$.

---

**Algorithm 1:** $\dot{\mathbb{H}}^1$-Gradient Ascent Algorithm

---

Initialize $\varphi^1$;
**for** $t = 1, 2, \cdots, T - 1$ **do**
$\quad \widehat{\varphi}^{t+1} = \varphi^t + \eta_t \boldsymbol{\nabla} \mathcal{I}_\nu^\mu(\varphi^t)$;
$\quad \varphi^{t+1} = (\widehat{\varphi}^{t+1})^{**}$;
**end**
**return** $\{\varphi^t\}_{t=1}^T$;

---

For an arbitrary function $\varphi$, its second convex conjugate $\varphi^{**}$ is always convex and satisfies $\varphi^{**} \leq \varphi$. Consequently, the step $\varphi^{t+1} = (\widehat{\varphi}^{t+1})^{**}$ can be interpreted as projecting $\widehat{\varphi}^{t+1}$ onto the space of convex functions. In addition, it holds that $\mathcal{I}_\nu^\mu(\varphi) \leq \mathcal{I}_\nu^\mu(\varphi^{**})$, indicating that applying the second convex conjugate does not reduce the functional value.

### 2.3 WASSERSTEIN GRADIENT

In this subsection, we review the definition of Wasserstein gradient and a Wasserstein gradient descent approach for finding the Wasserstein barycenter of absolutely continuous probability measures. Let $\mathcal{H} : \mathcal{P}_2^r(\Omega) \to \mathbb{R}$ be a functional over the nonlinear space $\mathcal{P}_2^r(\Omega)$. For any $\nu \in \mathcal{P}^r(\Omega) \cap L^\infty(\Omega)$, i.e., $\nu$ is absolutely continuous with an $L^\infty$ density function, we can define the first variation of $\mathcal{H}$. The map $\frac{\delta \mathcal{H}}{\delta \mu}(\mu) : \Omega \to \mathbb{R}$ is called the first variation of $\mathcal{H}$ at $\mu$, if

$$\frac{\mathrm{d}}{\mathrm{d}\epsilon} \mathcal{H}(\mu + \epsilon \chi) \bigg|_{\epsilon=0} = \int_\Omega \frac{\delta \mathcal{H}}{\delta \mu}(\mu)(x) \, \mathrm{d}\chi(x),$$

for the direction $\chi = \nu - \mu$. Lemma 10.4.1 (Ambrosio et al., 2008) implies that the Wasserstein gradient of $\mathcal{H}$ at $\mu$ is given by $\mathbb{W}\mathcal{H}(\mu) := \nabla \frac{\delta \mathcal{H}}{\delta \mu}(\mu)$ under mild conditions.

We remark here that the Wasserstein gradient is fundamentally different from the gradient in a Hilbert space as defined in Subsection 2.2. The primary reason is that $\mathcal{P}_2^r(\Omega)$ is not a linear vector space, and standard arithmetic operations such as addition and subtraction do not exist. For instance, given $\nu, \mu \in \mathcal{P}_2^r(\Omega)$, their difference $\nu - \mu$ is not a valid probability measure and hence $\nu - \mu \notin \mathcal{P}_2^r(\Omega)$. For the same reason, a different notion of convexity is appropriate for $\mathcal{H} : \mathcal{P}_2^r(\Omega) \to \mathbb{R}$. Specifically, $\mathcal{H}$ is said to be geodesically convex if, for any $\nu, \mu \in \mathcal{P}_2^r(\Omega)$ and $\epsilon \in [0, 1]$, it holds that $\mathcal{H}((\epsilon T_\nu^\mu + (1 - \epsilon) \operatorname{id})_\# \nu) \leq \epsilon \mathcal{H}(\mu) + (1 - \epsilon) \mathcal{H}(\nu)$.

## 3 NONCONVEX-CONCAVE MINIMAX FORMULATION FOR OPTIMAL TRANSPORT BARYCENTER

In this section, we formulate the Wasserstein barycenter problem as a nonconvex-concave optimization problem. By reviewing existing methods for computing the Wasserstein barycenter, we demonstrate that our nonconvex-concave formulation is more realistic and practical. We then propose a gradient descent-ascent type algorithm and provide relevant convergence analysis.

### 3.1 NONCONVEX-CONCAVE MINIMAX OPTIMIZATION IN EUCLIDEAN SPACE

Before presenting our algorithm, we first discuss nonconvex-concave optimization algorithms in Euclidean space to better understand the challenges and feasible objectives in such problems. Given a smooth function $f : \mathbb{R}^{d_1} \times \mathbb{R}^{d_2} \to \mathbb{R}$, the nonconvex-concave minimax optimization problem is generally formulated as

$$\min_{x \in \mathbb{R}^{d_1}} \max_{y \in \mathbb{Y}} f(x, y),$$

where $\mathbb{Y} \subset \mathbb{R}^{d_2}$ is convex and compact, and $f(x, \cdot)$ is concave for any fixed $x$ while $f(\cdot, y)$ can be nonconvex for a given $y$. Let $\Phi(x) := \max_{y \in \mathbb{Y}} f(x, y)$. If there exists a unique $y_x^* \in \mathbb{Y}$ that attains this maximal value, i.e., $\Phi(x) = f(x, y_x^*)$, then by Danskin's Theorem (Bernhard & Rapaport, 1995;

Bertsekas, 1997), $\Phi(x)$ is differentiable and the gradient can be computed as $\nabla \Phi(x) = \nabla_x f(x, y_x^*)$, where $\nabla_x$ computes the gradient with respect to $x$ only.

The ultimate goal of the minimax optimization problem is to find the global minimum of $\Phi$. However, such problem is NP-hard due to the nonconvexity of $\Phi$ (Lin et al., 2020). A common surrogate in nonconvex optimization is to seek a stationary point $x$ of $\Phi$, where $\nabla \Phi(x) = 0$. A simple and efficient method is the gradient descent-ascent (GDA) algorithm (Algorithm 2), where $\mathcal{P}_{\mathbb{Y}}$ is the projection operator onto $\mathbb{Y}$.

---

**Algorithm 2:** Gradient Descent-Ascent Algorithm on Euclidean Domain

Initialize $x_1, y_1$;
**for** $t = 1, 2, \cdots, T - 1$ **do**
$\quad x^{t+1} = x^t - \eta \nabla_x f(x^t, y^t)$;
$\quad y^{t+1} = \mathcal{P}_{\mathbb{Y}}(y^t + \tau \nabla_y f(x^t, y^t))$;
**end**
**return** $\{x^t, y^t\}_{t=1}^T$;

---

Despite the complex structure of the minimax problem and the nonconvexity of $\Phi$, the GDA algorithm remains theoretically trackable. Lin et al. (2020) proved that, with suitable choices of step sizes $(\eta, \tau)$, the following bound holds: $\min_{t \in [T]} \|\nabla \Phi(x^t)\|_2^2 \leq \frac{1}{T} \sum_{t=1}^T \|\nabla \Phi(x^t)\|_2^2 = O\left(\frac{1}{T}\right)$, which indicates that a good approximation of the stationary point can be achieved with $\varepsilon$-accuracy within the first $O(1/\varepsilon)$ iterations.

### 3.2 Existing Approach for Wasserstein barycenter

Given $n$ probability measures $\mu_1, \mu_2, \ldots, \mu_n \in \mathcal{P}_2^r(\Omega)$, the Wasserstein barycenter is defined as the minimizer of the barycenter functional $\mathcal{F} : \mathcal{P}_2^r(\Omega) \to \mathbb{R}$, given by

$$\mathcal{F}(\nu) := \frac{1}{n} \sum_{i=1}^n \mathcal{W}_2^2(\nu, \mu_i). \tag{3}$$

Wasserstein barycenter can be viewed as a generalization of the arithmetic mean in the Wasserstein space with metric $\mathcal{W}_2$. It is shown that the barycenter functional admits a Wasserstein gradient $\mathbb{W}\mathcal{F}(\nu) = \mathrm{id} - \frac{1}{n} \sum_{i=1}^n T_\nu^{\mu_i}$, and a Wasserstein gradient based approach has been proposed (Zemel & Panaretos, 2019; Chewi et al., 2020) for numerically computing the Wasserstein barycenter by iteratively updating as follows:

$$\nu^{t+1} = \left( \mathrm{id} - \eta_t \mathbb{W}\mathcal{F}(\nu^t) \right)_{\#} \nu^t.$$

However, the above algorithm implicitly assumes that the optimal transport maps $\{T_\nu^{\mu_i}\}_{i=1}^n$ are known. In practice, computing $T_\nu^{\mu_i}$ for multivariate distributions is particularly challenging and often can only be approximated to a certain accuracy, for example, by using the Sinkhorn algorithm.

In this work, we reformulate the Wasserstein barycenter problem as a nonconvex-concave minimax problem. Rather than computing the optimal transport maps in each iteration, we propose a gradient descent-ascent algorithm to solve the associated minimax problem, where the transport maps are updated using $\dot{\mathbb{H}}^1$ ascent in each iteration. Our approach alleviates the computational burden of solving $n$ optimal transport problems per iteration compared with the traditional approaches.

### 3.3 Our Approach: Wasserstein-Descent $\dot{\mathbb{H}}^1$-Ascent Algorithm

Let $\mathbb{F}_{\alpha,\beta}$ be a subset of $\dot{\mathbb{H}}^1$ consisting of all functions that are $\alpha$-strongly convex and $\beta$-smooth, i.e. for every $f \in \mathbb{F}_{\alpha,\beta}$ and $x, y \in \Omega$, it holds that $f \in \dot{\mathbb{H}}^1$ and $\frac{\alpha}{2}\|x-y\|_2^2 \leq f(x) - f(y) - \langle \nabla f(y), x - y \rangle \leq \frac{\beta}{2}\|x-y\|_2^2$. With this notation, $\mathbb{F}_{0,\infty}$ represents the set of all convex functions on $\Omega$. The dual formulation of Kantorovich problem implies

$$\mathcal{W}_2^2(\nu, \mu_i) = \max_{\varphi_i \in \mathbb{F}_{0,\infty}} \left\{ \mathcal{I}_\nu^{\mu_i}(\varphi_i) = \int_\Omega \frac{\|x\|_2^2}{2} - \varphi_i(x) \, \mathrm{d}\nu + \int_\Omega \frac{\|x\|_2^2}{2} - \varphi_i^* \, \mathrm{d}\mu_i(x) \right\}.$$

Given $n$ probability measures $\mu_1, \ldots, \mu_n \in \mathcal{P}_2^r(\Omega)$, we can reformulate the Wasserstein barycenter problem as

$$\min_{\nu \in \mathcal{P}_2^r(\Omega)} \max_{\varphi_i \in \mathbb{F}_{\alpha,\beta}} \left\{ \mathcal{J}(\nu, \boldsymbol{\varphi}) := \frac{1}{n} \sum_{i=1}^n \mathcal{I}_\nu^{\mu_i}(\varphi_i) \right\}. \tag{4}$$

Since the inner maximization part of (4) consists of $n$ separable subproblems, using the notation

$$\mathcal{L}^{\mu_i}(\nu) := \max_{\varphi_i \in \mathbb{F}_{\alpha,\beta}} \mathcal{I}_\nu^{\mu_i}(\varphi_i), \tag{5}$$

equation 4 can be rewritten as $\min_\nu \max_{\varphi_i \in \mathbb{F}_{\alpha,\beta}} \mathcal{J}(\nu, \boldsymbol{\varphi}) = \min_\nu \frac{1}{n} \sum_{i=1}^n \mathcal{L}^{\mu_i}(\nu)$. When $\alpha = 0$ and $\beta = \infty$, $\mathbb{F}_{\alpha,\beta}$ is the set of convex functions and $\mathcal{W}_2^2(\nu, \mu) = \max_{\varphi \in \mathbb{F}_{\alpha,\beta}} \mathcal{I}_\nu^\mu(\varphi)$. Thus, we have $\min_\nu \frac{1}{n} \sum_{i=1}^n \mathcal{L}^{\mu_i}(\nu) = \min_\nu \mathcal{F}(\nu)$. The constraint set $\mathbb{F}_{\alpha,\beta}$ enforces additional regularity on the Kantorovich potentials. This technique has been frequently used in the optimal transport literature (Paty et al., 2020; Hütter & Rigollet, 2021; Manole et al., 2024).

Fix $\nu$, the objective functional $\mathcal{J}(\nu, \boldsymbol{\varphi})$ is concave in each $\varphi_i$. However, if we fix $\{\varphi_i\}_{i=1}^n \subset \mathbb{F}_{\alpha,\beta}$ without further assumptions, $\mathcal{J}(\nu, \boldsymbol{\varphi})$ is not geodesically convex unless $\beta \leq 1$. Thus, problem 4 is a "nonconvex-concave" minimax optimization problem.

We now discuss different notions of gradients for the objective functional $\mathcal{J} : \mathcal{P}_2^r(\Omega) \times \dot{\mathbb{H}}^1 \times \cdots \times \dot{\mathbb{H}}^1 \to \mathbb{R}$. Given $\nu \in \mathcal{P}_2^t(\Omega)$, the $\dot{\mathbb{H}}^1$ gradient of $\mathcal{J}$ with respect to $\varphi_i$ can be computed using equation 2. For a fixed set $\{\varphi_i\}_{i=1}^n \subset \mathbb{F}_{\alpha,\beta}$, the definitions in subsection 2.3 imply that the Wasserstein gradient of $\mathcal{J}$ is given by $\mathbb{W}\mathcal{J}(\nu, \boldsymbol{\varphi}) = \mathrm{id} - \nabla\overline{\varphi}$, where $\overline{\varphi} = \frac{1}{n} \sum_{i=1}^n \varphi_i$. Before introducing a GDA type algorithm for solving the minimax optimization in equation 4, we summarize different notions of gradients for readers' convenience:

- the usual gradient of $\varphi_i : \mathbb{R}^d \to \mathbb{R}$ is denoted as $\nabla\varphi_i$;
- the $\dot{\mathbb{H}}^1$ gradient of $\mathcal{J}$ with respect to $\varphi_i$ is computed as $\boldsymbol{\nabla}_{\varphi_i} \mathcal{J}(\nu, \boldsymbol{\varphi}) = \frac{1}{n}(-\Delta)^{-1}(-\nu + (\nabla\varphi_i^*)_\# \mu_i)$;
- the Wasserstein gradient of $\mathcal{J}$ with respect to $\nu$ is computed as $\mathbb{W}\mathcal{J}(\nu, \boldsymbol{\varphi}) = \mathrm{id} - \nabla\overline{\varphi}$, where $\overline{\varphi} = \frac{1}{n} \sum_{i=1}^n \varphi_i$.

Let $\mathcal{P}_{\mathbb{F}_{\alpha,\beta}}$ be the projection operator onto $\mathbb{F}_{\alpha,\beta}$. This projection is well-defined and unique since $\mathbb{F}_{\alpha,\beta} \subset \dot{\mathbb{H}}^1$ is a complete and convex metric space. We propose the following *Wasserstein-Descent* $\dot{\mathbb{H}}^1$-*Ascent* (WDHA) algorithm, of which the pseudocode is provided in Algorithm 3.

---

**Algorithm 3:** Wasserstein-Descent $\dot{\mathbb{H}}^1$-Ascent Algorithm

---

Initialize $\nu^1, \boldsymbol{\varphi}^1$;
**for** $t = 1, 2, \cdots, T - 1$ **do**
    **for** $i = 1, 2, \ldots, n$ **do**
        $\widehat{\varphi}_i^{t+1} = \varphi_i^t + \eta \boldsymbol{\nabla}_{\varphi_i} \mathcal{J}(\nu^t, \boldsymbol{\varphi}^t)$;
        $\varphi_i^{t+1} = \mathcal{P}_{\mathbb{F}_{\alpha,\beta}}(\widehat{\varphi}_i^{t+1})$;
    **end**
    $\nu^{t+1} = (\mathrm{id} - \tau \mathbb{W}\mathcal{J}(\nu^t, \boldsymbol{\varphi}^t))_\# \nu^t$;
**end**
**return** $\{(\nu^t, \boldsymbol{\varphi}^t)\}_{t=1}^T$;

---

Since $\mathcal{J}(\cdot, \boldsymbol{\varphi})$ is a functional on the space of absolutely continuous probability measures, it can also be viewed as a functional mapping density functions to $\mathbb{R}$. By embedding all densities functions into $L^2(\Omega)$, we can alternatively use the $L^2$-gradient to update $\nu$. However, in simulations, the Wasserstein gradient update significantly outperforms the $L^2$-gradient update.

## 3.4 CONVERGENCE ANALYSIS

We now establish the notion of stationary points for $\mathcal{F}_{\alpha,\beta}(\nu) := \frac{1}{n} \sum_{i=1}^n \mathcal{L}^{\mu_i}(\nu)$ and show that the output sequence from Algorithm 3 converges to a stationary point of $\mathcal{F}_{\alpha,\beta}(\nu)$. We start from presenting several properties of the functionals $\mathcal{I}_{\alpha,\beta}$ and $\mathcal{L}^\mu$.

The first lemma demonstrates the strong concavity and smoothness of the functional $\mathcal{I}_\nu^\mu$ on $\mathbb{F}_{\alpha,\beta}$ with $0 < \alpha \le \beta < \infty$, when the density function of $\mu$ is bounded from below and above.

**Lemma 1** (Strong concavity and smoothness of $\mathcal{I}_\nu^\mu$). *If $0 < a \le \mu(x) \le b < \infty$ for all $x \in \Omega$, then for any $\varphi_1, \varphi_2 \in \mathbb{F}_{\alpha,\beta}$, set $A = a\alpha^d/\beta$ and $B = b\beta^d/\alpha$, the following inequalities hold,*

$$-\frac{A}{2}\|\varphi_2 - \varphi_1\|_{\dot{\mathbb{H}}^1}^2 \ge \mathcal{I}_\nu^\mu(\varphi_2) - \mathcal{I}_\nu^\mu(\varphi_1) - \langle \boldsymbol{\nabla}\mathcal{I}_\nu^\mu(\varphi_1), \varphi_2 - \varphi_1 \rangle_{\dot{\mathbb{H}}^1} \ge -\frac{B}{2}\|\varphi_2 - \varphi_1\|_{\dot{\mathbb{H}}^1}^2.$$

The following lemma provides an explicit form of the Wasserstein gradient of $\mathcal{L}^\mu$.

**Lemma 2.** *If $0 < a \le \mu(x) \le b < \infty$ for all $x \in \Omega$, then $\mathcal{I}_\nu^\mu(\varphi)$ admits a unique maximizer in $\mathbb{F}_{\alpha,\beta}$. Let $\widetilde{\varphi}_\nu^\mu := \arg\max_{\varphi \in \mathbb{F}_{\alpha,\beta}} \mathcal{I}_\nu^\mu(\varphi)$. Then, we have*

- *the first variation of $\mathcal{L}^\mu$ at $\nu$ is $\frac{\delta \mathcal{L}^\mu}{\delta \nu}(\nu) = \frac{\langle \mathrm{id}, \mathrm{id} \rangle}{2} - \widetilde{\varphi}_\nu^\mu$;*

- *the Wasserstein gradient of $\mathcal{L}^\mu$ at $\nu$ is $\mathbb{W}\mathcal{L}^\mu(\nu) = \mathrm{id} - \nabla\widetilde{\varphi}_\nu^\mu$.*

The above result directly implies that $\mathbb{W}\mathcal{F}_{\alpha,\beta}(\nu) = \mathrm{id} - \nabla\overline{\widetilde{\varphi}}_\nu^\mu$, where $\overline{\widetilde{\varphi}}_\nu^\mu = \frac{1}{n}\sum_{i=1}^n \widetilde{\varphi}_\nu^{\mu_i}$. Consequently, it is natural to define the stationary points of $\mathcal{F}_{\alpha,\beta}$ as probability measures for which the Wasserstein gradient has zero $L^2$-norm.

**Definition 1.** *We call $\nu \in \mathcal{P}_2^r(\Omega)$ a stationary point of $\mathcal{F}_{\alpha,\beta}$ if and only if $\int_\Omega \|\mathbb{W}\mathcal{F}_{\alpha,\beta}(\nu)\|_2^2 \, \mathrm{d}\nu = 0$.*

We denote the dual norm of $\|\cdot\|_{\dot{\mathbb{H}}^1}$ as $\|\cdot\|_{\dot{\mathbb{H}}^{-1}}$, defined by $\|\nu\|_{\dot{\mathbb{H}}^{-1}} := \inf\{\int_\Omega \varphi \, \mathrm{d}\nu \mid \|\varphi\|_{\dot{\mathbb{H}}^1} \le 1\}$. For more information on $\|\cdot\|_{\dot{\mathbb{H}}^{-1}}$, we refer readers to Chapter 5 of (Santambrogio, 2015). The following lemma indicates that the Wasserstein gradient $\mathbb{W}\mathcal{L}^\mu$ is Lipschitz continuous with constant $1/A$ with respect to the $\dot{\mathbb{H}}^{-1}$-norm, where $A$ is the constant from Lemma 1.

**Lemma 3** (Lipschitzness of Wasserstein gradient $\mathbb{W}\mathcal{L}^\mu$). *If $0 < a \le \mu(x) \le b < \infty$, then*

$$\|\mathbb{W}\mathcal{L}^\mu(\nu_1) - \mathbb{W}\mathcal{L}^\mu(\nu_2)\|_{L^2} = \|\widetilde{\varphi}_{\nu_2}^\mu - \widetilde{\varphi}_{\nu_1}^\mu\|_{\dot{\mathbb{H}}^1} \le A^{-1}\|\nu_1 - \nu_2\|_{\dot{\mathbb{H}}^{-1}},$$

*where $\|\cdot\|_{L^2}$ denote the $L^2$-norm of the function.*

Applying Theorem 5.34 in (Santambrogio, 2015), the above result further implies

$$A\|\widetilde{\varphi}_{\nu_2}^\mu - \widetilde{\varphi}_{\nu_1}^\mu\|_{\dot{\mathbb{H}}^1} \le \|\nu_1 - \nu_2\|_{\dot{\mathbb{H}}^{-1}} \le \sqrt{\max\{\|\nu_1\|_\infty, \|\nu_2\|_\infty\}} \mathcal{W}_2(\nu_1, \nu_2). \tag{6}$$

We emphasize that the inequality above holds because $\widetilde{\varphi}_{\nu_2}^\mu, \widetilde{\varphi}_{\nu_1}^\mu$ are restricted to $\mathbb{F}_{\alpha,\beta}$. Otherwise, only a weaker bound in $\mathcal{W}_1$ metric is available (Theorem 1.3, Berman, 2021), $\|\varphi_{\nu_2}^\mu - \varphi_{\nu_1}^\mu\|_{\dot{\mathbb{H}}^1} \le c_1\sqrt{\mathcal{W}_1(\nu_1, \nu_2)}$, where $\mathcal{W}_1(\nu_1, \nu_2)$ is the 1-Wasserstein distance between $\nu_1$ and $\nu_2$, and $c_1$ is a constant depending on $\nu_1$ and $\nu_2$. Following standard notations in the literature, we define $\|\nu_1\|_\infty = \sup_x \nu_1(x)$ and $\|\nu_2\|_\infty = \sup_x \nu_2(x)$, where $\nu_1(x)$ and $\nu_2(x)$ are density functions of $\nu_1$ and $\nu_2$ evaluated at the point $x \in \Omega$.

Following the above discussion and applying Lemma 3, we derive the smoothness of $\mathcal{L}^\mu$ with respect to the $\mathcal{W}_2$ metric.

**Lemma 4** (Smoothness of $\mathcal{L}^\mu$). *Let $C = 1 + \alpha + \frac{\max\{\|\nu_1\|_\infty, \|\nu_2\|_\infty\}}{A}$, we have*

$$\mathcal{L}^\mu(\nu_2) - \mathcal{L}^\mu(\nu_1) \le \int_\Omega \langle \mathrm{id} - \nabla\widetilde{\varphi}_{\nu_1}^\mu, T_{\nu_1}^{\nu_2} - \mathrm{id} \rangle \, \mathrm{d}\nu_1 + \frac{C}{2}\mathcal{W}_2^2(\nu_1, \nu_2).$$

Finally, we establish the convergence of the WDHA algorithm 3 to a stationary point of $\mathcal{F}_{\alpha,\beta}$ in the following theorem.

**Theorem 1** (Convergence rate of WDHA). *Assume that there are constant $a$ and $b$, such that the density functions satisfy $0 < a \le \mu_i(x) \le b < \infty$ for all $i = 1, 2, \ldots, n$ and $x \in \Omega$. Recall that $A = a\alpha^d/\beta$ and $B = b\beta^d/\alpha$. If $\max_t \|\nu^t\|_\infty \le V < \infty$ for some constant $V > 0$, by choosing the step sizes $(\tau, \eta)$ satisfying $\eta < 1/B$ and $\tau < \frac{A^2\eta}{A\eta(A\alpha + A + V) + 4V\sqrt{4 - 2A\eta}}$, we have*

$$\min_{t \in [T]} \int_\Omega \|\mathbb{W}\mathcal{F}_{\alpha,\beta}(\nu^t)\|_2^2 \, \mathrm{d}\nu^t$$

$$\le \frac{1}{T}\sum_{t=1}^T \int_\Omega \|\mathbb{W}\mathcal{F}_{\alpha,\beta}(\nu^t)\|_2^2 \, \mathrm{d}\nu^t \le \frac{\frac{4\tau V\bar{\delta}^1}{A\eta} + \mathcal{F}_{\alpha,\beta}(\nu^1) - \mathcal{F}_{\alpha,\beta}(\nu^{T+1})}{T\tau/2},$$

*where $\bar{\delta}^1 = \bar{\delta}^1(\nu^1, \boldsymbol{\varphi}^1, \mu_1, \ldots, \mu_n) > 0$ is a constant.*

**Remark 1.** *(i) The minimum value of the squared $L^2$-norm of the Wasserstein gradient $\mathbb{W}\mathcal{F}_{\alpha,\beta}$ over the first $T$ iterations converges to zero at a rate of $O(T^{-1})$. This convergence rate is consistent with the results obtained by using GDA to solve nonconvex-concave minimax problems in the Euclidean space, as demonstrated by* Lin et al. (2020). *(ii) We assume that the density functions of all iterates $\nu^t$ are uniformly bounded. In practice, we have not encountered any case where $\|\nu^t\|_\infty$ diverges. Therefore, we hypothesize that this technical assumption can be inferred from other assumptions, which we leave as an open problem. (iii) By definition, $\overline{\nu}$ is a Wasserstein barycenter if $id - \frac{1}{n}\sum_{i=1}^n \nabla\varphi_{\overline{\nu}}^{\mu_i} = 0$, where $\varphi_{\overline{\nu}}^{\mu_i}$ is the Kantorovich potential between $\overline{\nu}$ and $\mu_i$. If $\varphi_{\overline{\nu}}^{\mu_i} \in \mathbb{F}_{\alpha,\beta}$ for all $i$, then $\overline{\nu}$ is a stationary point of $\mathcal{F}_{\alpha,\beta}$, i.e., $\mathbb{W}\mathcal{F}_{\alpha,\beta}(\overline{\nu}) = 0$. Reversely, if we assume that the Kantorovich potential between the true barycenter and each $\mu_i$ is in $\mathbb{F}_{\alpha,\beta}$, then $\mathbb{W}\mathcal{F}_{\alpha,\beta}(\overline{\nu}') = 0$ would mean that $\overline{\nu}'$ is a Wasserstein barycenter.*

### 3.5 COMPUTATIONAL COMPLEXITIES

In this subsection, we discuss the implementation and computational complexity of the WDHA algorithm (Algorithm 3). To implement Algorithm 3, we need to numerically approximate the infinite-dimensional objects $\nu, \varphi_1, \ldots, \varphi_n$ through discretization. Given $\nu$ and $\varphi$ supported on a fixed grid of size $m$, the computation of the convex conjugate $\varphi^*$, the pushforward measure $(\nabla\varphi)_\#\nu$, and the negative inverse Laplacian $(-\Delta)^{-1}(\nu)$ with zero Neumann boundary conditions only requires a time complexity of $O(m\log(m))$ and space complexity of $O(m)$, as demonstrated by Jacobs & Léger (2020). However, computing the projection $\mathcal{P}_{\mathbb{F}_{\alpha,\beta}}(\varphi)$ is computationally expensive, with a time complexity $O(m^2)$ (Simonetto, 2021). For more efficient computation, we recommend replacing the projection step $\mathcal{P}_{\mathbb{F}_{\alpha,\beta}}(\varphi)$ with computing the second convex conjugate $(\varphi)^{**}$ in Algorithm 3 in practice. Although $(\varphi)^{**}$ only enforces the convexity and not strong convexity or smoothness, the modified algorithm performs well empirically. This adjusted algorithm achieves a time complexity $O(m\log(m))$ per iteration, and the pseudocode is provided below.

---

**Algorithm 4:** Wasserstein-Descent $\dot{\mathbb{H}}^1$-Ascent Algorithm

Initialize $\nu^1, \boldsymbol{\varphi}^1$;
**for** $t = 1, 2, \cdots, T-1$ **do**
    **for** $i = 1, 2, \ldots, n$ **do**
        $\widehat{\varphi}_i^{t+1} = \varphi_i^t + \eta_i^t \boldsymbol{\nabla}_{\varphi_i}\mathcal{J}(\nu^t, \boldsymbol{\varphi}^t)$;
        $\varphi_i^{t+1} = (\widehat{\varphi}_i^{t+1})^{**}$;
    **end**
    $\nu^{t+1} = (id - \tau_t \mathbb{W}\mathcal{J}(\nu^t, \boldsymbol{\varphi}^t))_\# \nu^t$;
**end**
**return** $\{(\nu^t, \boldsymbol{\varphi}^t)\}_{t=1}^T$;

---

Theorem 1 suggests that the parameters $\tau, \eta$ should be bounded above. Empirically, we find that the above algorithm works better with diminishing step sizes, potentially due to two reasons: (1) diminishing step sizes may reduce the effect of discrete approximations; and (2) diminishing step sizes may be more effective for nonsmooth convex functions. Since the second convex conjugate does not enforce strong convexity and smoothness, Lemma 1 no longer holds, and $\mathcal{I}_\nu^\mu$ is only guaranteed to be concave. In addition, diminishing step size may speed up the learning process in the early stages and a inverse time decay for $\tau_t$, i.e., $\tau_t = 1/t$, works equally well in the simulation studies.

## 4 NUMERICAL STUDIES

Using both synthetic and real data, we compare our approach with two existing methods applicable to distributions supported on large grids: (1) Convolutional Wasserstein Barycenter (CWB) (Solomon et al., 2015) and (2) Debiased Sinkhorn Barycenter (DSB) (Janati et al., 2020). Both CWB and DSB employ entropic regularization techniques and are implemented as Python functions `convolutional_barycenter2d` and `convolutional_barycenter2d_debiased`, respectively, in the Python library "POT: Python Optimal Transport" (Flamary et al., 2021).

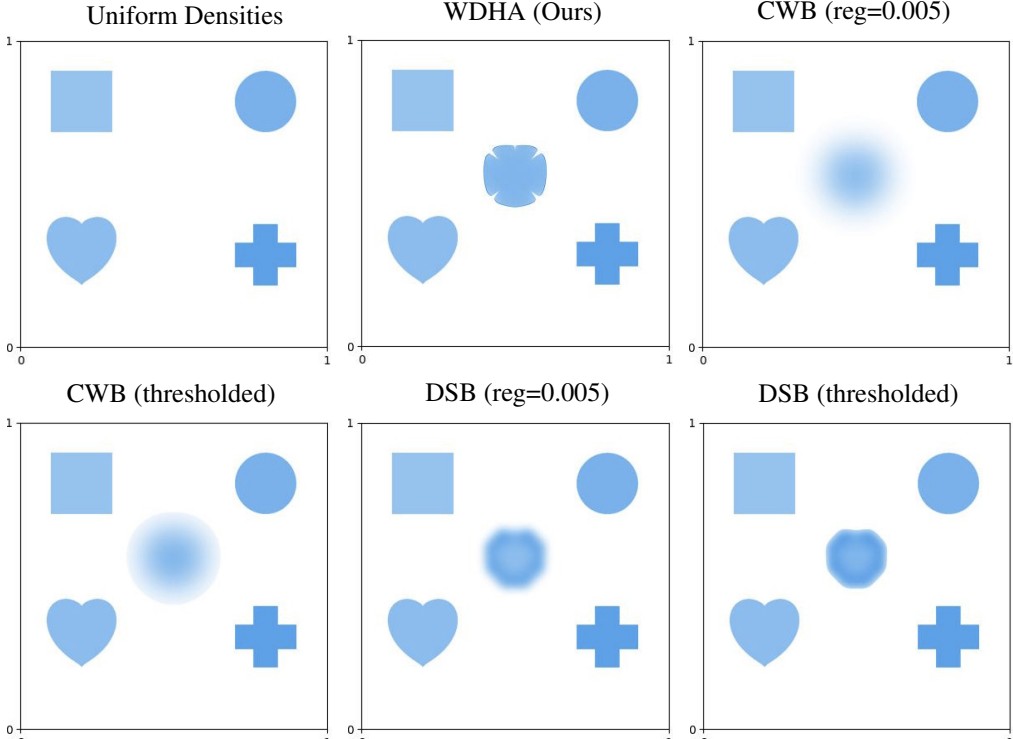

Figure 1: Illustration of Wasserstein barycenters computed using different methods. The goal is to compute the barycenter of four uniform densities supported on the square, circle, heart, and cross, respectively, as displayed in the top left image. The blended shape shown in the top middle image is the barycentric density computed using our method. Barycentric densities computed using CWB and DSB with regularization strength parameter reg = 0.005, and their thresholded versions are shown in the top right image and the bottom three images.

## 4.1 SYNTHETIC UNIFORM DISTRIBUTIONS

In this example, we aim to compute the barycenter of four uniform distributions whose supports are contained in $[0, 1]^2$ and take the shapes of a square, a circle, a heart, and a cross, respectively. Their densities are discretized on a fixed grid of size $m = 1024 \times 1024$ and are displayed in Figure 1 (top left). We apply Algorithm 4 and set $\tau_t = \exp(-t/T)$ and $\eta_i^1 = 0.05$ for all $i$ and decrease it by a factor of 0.99 if $\mathcal{I}_{\nu^t}^{\mu_i}(\varphi_i^{t+1}) < \mathcal{I}_{\nu^t}^{\mu_i}(\varphi_i^t)$.

The barycenters computed by our method, CWB, and DSB after 300 iterations are displayed in Figure 1. The density of the barycenter distribution generated by our method, as shown in the center of the top middle image in Figure 1, is uniformly distributed over a blended shape comprising a square, a circle, a heart, and a cross, with sharp edges. In contrast, the barycenters computed using CWB with regularization strength parameter reg = 0.005, displayed in the top right image in Figure 1, appear blurred. DSB yields a better representation than CWB but remains unclear; see the bottom middle image in Figure 1 for barycenters produced by DSB with reg = 0.005. Notably, we encounter a division by zero error if the regularization strength parameter is set to 0.001. We also compute thresholded versions of barycenters of CWB and DSB by removing intensities smaller than the threshold such that the removed intensities amount to $10\%$ of the total mass. The thresholded barycenters are shown in the bottom left and right images in Figure 1. However, the resulting barycenters lack the inward sharp curvature inherited from the heart and cross shape. Thus, the barycenter obtained by our method offers a clearer and more representative summary of the set.

We report the program run times for computing these barycenters and the corresponding 2-Wasserstein barycenter functional values $\mathcal{F}(\nu^{est})$, where $\nu^{est}$ represents the computed barycenter. All functional values reported below are estimated using the back-and-forth approach (Jacobs & Léger, 2020) and are multiplied by $10^3$. All methods were executed on Google Colab with an L4

Wasserstein Barycenter of Handwritten Eight

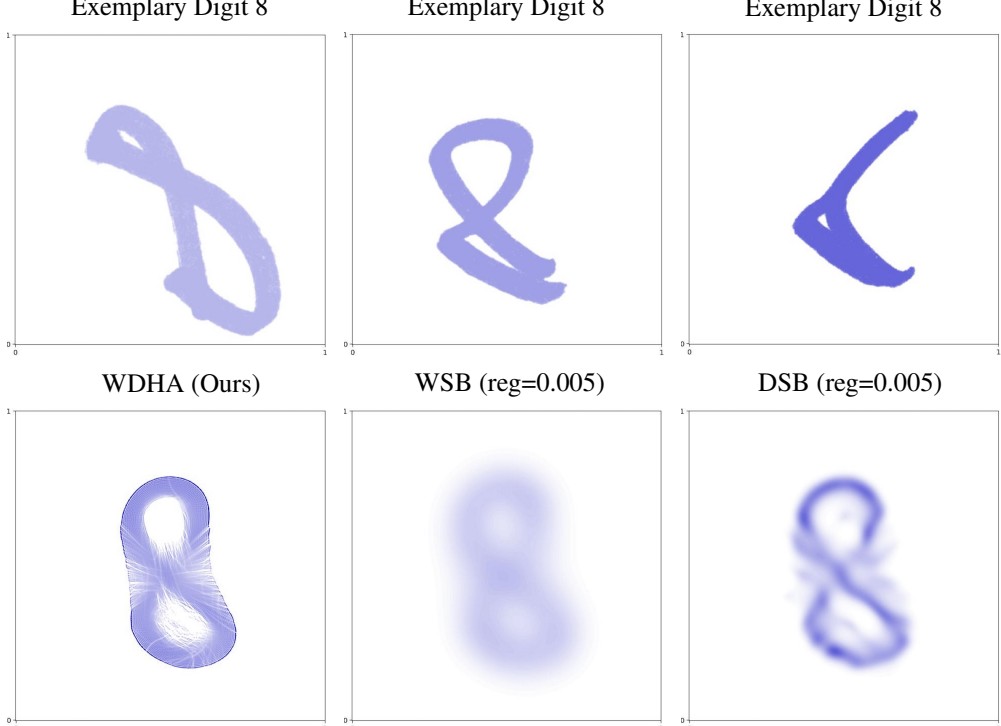

Figure 2: Top row displays three exemplary digit 8 images. Bottom row displays barycenters computed by different methods using 300 iterations.

GPU. Our method takes 676 seconds, whereas CWB takes 3731 seconds and DSB takes 7249 seconds. Additionally, our method achieves the smallest barycenter functional value (74.5791), compared to CWB and DSB, which yield values of 75.0689 and 74.5804, respectively. The functional values for the thresholded barycenters are 74.7346 (CWB) and 74.5921 (DSB).

## 4.2 HIGH-RESOLUTION HANDWRITTEN DIGITS

Here, our method is applied to the high-resolution handwritten digits data (Beaulac & Rosenthal, 2022). By treating the digit images as densities, we aim to compute the barycenter of one hundred handwritten images of the digit 8, each with a size of $500 \times 500$ pixels. Three exemplary images are displayed in the top row of Figure 2. To run Algorithm 4, we set $\tau_t = \exp(-t/T)$, and $\eta_i^1 = 0.5$ at iteration $t = 1$ and decrease it by a factor of 0.95 whenever $\mathcal{I}_{\nu^t}^{\mu_i}(\varphi_i^{t+1}) < \mathcal{I}_{\nu^t}^{\mu_i}(\varphi_i^t)$. The barycenters computed by our method, CWB, and DSB using $T = 300$ iterations are displayed in the bottom row of Figure 2. The barycenter computed by WDHA exhibits clearer and more detailed textures, revealing variations of the digits viewed as densities in the Wasserstein space. Furthermore, our method is more time-efficient, taking 3,299 seconds compared to 10,808 seconds for CWB and 11,186 seconds for DSB.

## 5 CONCLUSION

In this paper, we introduced a Wasserstein-Descent $\dot{\mathbb{H}}^1$-Ascent (WDHA) algorithm for computing the Wasserstein barycenter of $n$ probability density functions supported on a compact subset of $\mathbb{R}^d$. Our key technique is motivated by the recent progress in nonconvex-concave minimax optimization problems in the Euclidean space. Compared to existing methods for high-resolution densities, the WDHA algorithm is computationally more efficient and produces a clearer, sharper, and more detailed barycenter. We believe that our work not only advances computational techniques for Wasserstein barycenters but also sheds new light on optimizing nonlinear functionals using a combination of geometric structures.

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

## A   LIST OF NOTATIONS

| Notations | Meaning |
|---|---|
| $\dot{\mathbb{H}}^1$ | homogeneous Sobolev space |
| $\dot{\mathbb{H}}^{-1}$ | dual space of $\dot{\mathbb{H}}^1$ |
| $\mathbb{F}_{\alpha,\beta}$ | a subset of $\dot{\mathbb{H}}^1$ consisting of $\alpha$-strongly convex and $\beta$ smooth functions |
| $\mathcal{P}_2(\Omega)$ | space of probability measures with finite second order moment |
| $\mathcal{P}_2^r(\Omega)$ | subset of $\mathcal{P}_2(\Omega)$ consisting of absolutely continuous probability measures |
| $T_{\#}\mu$ | pushforward measure of $\mu$ under the map $T$ |
| $T_\nu^\mu$ | the optimal transport map from $\nu$ to $\mu$ |
| $\mathcal{W}_p(\mu,\nu)$ | $p$-Wasserstein distance between $\mu$ and $\nu$ |
| $\mathcal{I}_\nu^\mu$ | Kantorovich dual functional defined in equation 1 |
| $\frac{\delta\mathcal{F}}{\delta\mu}$ | first variation of the functional $\mathcal{F}:\mathcal{P}_2^r(\Omega)\to\mathbb{R}$ |
| $\mathbb{W}\mathcal{F}$ | Wasserstein gradient of the functional $\mathcal{F}$ |
| $\mathcal{J}(\nu,\boldsymbol{\varphi})$ | Wasserstein barycenter functional defined in equation 4 |
| $\mathcal{L}^\mu(\nu)$ | the maximal functinoal defined in equation 5 |
| $\mathcal{F}_{\alpha,\beta}$ | average of maximal functionals defined as $\mathcal{F}_{\alpha,\beta}=\frac{1}{n}\sum_{i=1}^n\mathcal{L}^{\mu_i}$ |
| $(-\Delta)^{-1}$ | negative inverse Laplacian operator with zero Neumann boundary conditions |
| $\mathcal{P}_{\mathbb{F}_{\alpha,\beta}}$ | projection operator onto $\mathbb{F}_{\alpha,\beta}$ |
| $\widetilde{\varphi}_\nu^\mu$ | best $\alpha$-strongly convex, $\beta$-smooth Kantorovich potential defined in Lemma 2 |
| $\varphi^*$ | convex conjugate of the function $\varphi$ |
| $\nabla\varphi$ | (standard) gradient of the function $\varphi$ |
| id | identity map |
| $\|\varphi\|_{L^2}$ | $L^2$-norm of $\varphi$, defined as $(\int_\Omega\|\varphi\|_2^2\,\mathrm{d}x)^{1/2}$ |
| $\|\varphi\|_{L^2(\nu)}$ | $L^2(\nu)$-norm of $\varphi$, defined as $(\int_\Omega\|\varphi\|_2^2\,\mathrm{d}\nu)^{1/2}$ |

## B   TECHNICAL DETAILS

### B.1   PROOF OF LEMMA 1

Before proving the lemma, let us demonstrate a technical result for computing the $\dot{\mathbb{H}}^1$ inner product first.

**Lemma 5.** *For any functions $\varphi_1,\varphi_2\in\dot{\mathbb{H}}^1$ and $\mu,\nu\in\mathcal{P}_2^r(\Omega)$, we have*

$$\langle\boldsymbol{\nabla}\mathcal{I}_\nu^\mu(\varphi_1),\varphi_2-\varphi_1\rangle_{\dot{\mathbb{H}}^1}=\int_\Omega\varphi_1-\varphi_2\,\mathrm{d}\nu-\int_\Omega\left[\varphi_1\circ\nabla\varphi_1^*-\varphi_2\circ\nabla\varphi_1^*\right]\mathrm{d}\mu.$$

*Proof.* Let $g=\boldsymbol{\nabla}\mathcal{I}_\nu^\mu(\varphi_1)$. By the definition of $\boldsymbol{\nabla}\mathcal{I}_\nu^\mu$ in equation 2, we have

$$\Delta g=\nu-(\nabla\varphi_1^*)_{\#}\mu.$$

Therefore, we have

$$\langle\boldsymbol{\nabla}\mathcal{I}_\nu^\mu(\varphi_1),\varphi_2-\varphi_1\rangle_{\dot{\mathbb{H}}^1}\overset{(i)}{=}\int_\Omega\langle\nabla g,\nabla\varphi_2-\nabla\varphi_1\rangle\,\mathrm{d}x\overset{(ii)}{=}-\int_\Omega(\varphi_2-\varphi_1)\Delta g\,\mathrm{d}x$$

$$=\int_\Omega(\varphi_1-\varphi_2)\left[\nu-(\nabla\varphi_1^*)_{\#}\mu\right]\mathrm{d}x$$

$$=\int_\Omega\varphi_1-\varphi_2\,\mathrm{d}\nu-\int_\Omega\varphi_1\circ\nabla\varphi_1^*-\varphi_2\circ\nabla\varphi_1^*\,\mathrm{d}\mu.$$

Here, (i) is by the definition of inner product in $\dot{\mathbb{H}}^1$, and (ii) is due to integration by parts.   □

Let us now prove the lemma.

*Proof of Lemma 1.* Note that we have

$$
\mathcal{I}_\nu^\mu(\varphi_2) - \mathcal{I}_\nu^\mu(\varphi_1) - \langle \boldsymbol{\nabla}\mathcal{I}_\nu^\mu(\varphi_1), \varphi_2 - \varphi_1 \rangle_{\dot{\mathbb{H}}^1}
$$

$$
\stackrel{\text{(i)}}{=} \Big[ \int_\Omega \frac{\|x\|_2^2}{2} - \varphi_2(x)\,\mathrm{d}\nu(x) + \int_\Omega \frac{\|x\|_2^2}{2} - \varphi_2^*(x)\,\mathrm{d}\mu(x) \Big]
$$

$$
- \Big[ \int_\Omega \frac{\|x\|_2^2}{2} - \varphi_1(x)\,\mathrm{d}\nu(x) + \int_\Omega \frac{\|x\|_2^2}{2} - \varphi_1^*(x)\,\mathrm{d}\mu(x) \Big]
$$

$$
- \Big[ \int_\Omega \varphi_1(x) - \varphi_2(x)\,\mathrm{d}\nu(x) - \int_\Omega (\varphi_1 \circ \nabla\varphi_1^*)(x) - (\varphi_2 \circ \nabla\varphi_1^*)(x)\,\mathrm{d}\mu(x) \Big]
$$

$$
= \int_\Omega \varphi_1^*(x) - \varphi_2^*(x) + (\varphi_1 \circ \nabla\varphi_1^*)(x) - (\varphi_2 \circ \nabla\varphi_1^*)(x)\,\mathrm{d}\mu(x). \tag{7}
$$

Here, we use the definition of $\mathcal{I}_\nu^\mu$ and Lemma 5 to derive (i). By properties of convex conjugate, $\nabla\varphi^*(y) = \arg\max_{x\in\Omega}\langle x,y\rangle - \varphi(x)$, which further implies that

$$
\varphi^*(y) = \langle \nabla\varphi^*(y), y \rangle - \varphi(\nabla\varphi^*(y)).
$$

Combining the above quality with equation 7 yields

$$
\mathcal{I}_\nu^\mu(\varphi_2) - \mathcal{I}_\nu^\mu(\varphi_1) - \langle \boldsymbol{\nabla}\mathcal{I}_\nu^\mu(\varphi_1), \varphi_2 - \varphi_1 \rangle_{\dot{\mathbb{H}}^1}
$$

$$
= \int_\Omega \nabla\varphi_1^*(x)^\top x - \big[ \nabla\varphi_2^*(x)^\top x - \varphi_2\big(\nabla\varphi_2^*(x)\big) \big] - \varphi_2\big(\nabla\varphi_1^*(x)\big)\,\mathrm{d}\mu(x)
$$

$$
\stackrel{\text{(i)}}{=} -\int_\Omega \varphi_2\big(\nabla\varphi_1^*(x)\big) - \varphi_2\big(\nabla\varphi_2^*(x)\big) - \big\langle \nabla\varphi_2\big(\nabla\varphi_2^*(x)\big), \nabla\varphi_2^*(x) - \nabla\varphi_1^*(x) \big\rangle\,\mathrm{d}\mu(x)
$$

$$
= -\int_\Omega \mathcal{B}_{\varphi_2}\big(\nabla\varphi_1^*(x), \nabla\varphi_2^*(x)\big)\,\mathrm{d}\mu(x),
$$

where $\mathcal{B}_{\varphi_2}$ is the Bregman divergence of $\varphi_2$ defined as

$$
\mathcal{B}_{\varphi_2}(x, x') := \varphi_2(x) - \varphi_2(x') - \langle \nabla\varphi_2(x'), x - x' \rangle.
$$

Here, in (i), we use the fact that $\nabla\varphi_2 \circ \nabla\varphi_2^* = \mathrm{id}$. By properties of Bregman divergences,

$$
\mathcal{B}_{\varphi_2}(\nabla\varphi_1^*(x), \nabla\varphi_2^*(x)) = \mathcal{B}_{\varphi_2^*}(\nabla\varphi_2 \circ \nabla\varphi_1^*(x), \nabla\varphi_2 \circ \nabla\varphi_2^*(x)) = \mathcal{B}_{\varphi_2^*}(\nabla\varphi_2 \circ \nabla\varphi_1^*(x), x).
$$

Since $\varphi_2$ is $\alpha$-strongly convex and $\beta$-smooth, we know $\varphi_2^*$ is $1/\beta$-strongly convex and $1/\alpha$-smooth. Thus, we have

$$
\frac{1}{2\beta}\|\nabla\varphi_2 \circ \nabla\varphi_1^*(x) - x\|_2^2 \leq \mathcal{B}_{\varphi_2^*}(\nabla\varphi_2 \circ \nabla\varphi_1^*(x), x) \leq \frac{1}{2\alpha}\|\nabla\varphi_2 \circ \nabla\varphi_1^*(x) - x\|_2^2,
$$

Integrating the above inequality with respect to $\mu$ and applying change of variable formulas entail

$$
\frac{1}{2\beta}\int_\Omega \|\nabla\varphi_2 - \nabla\varphi_1(x)\|_2^2\,\mathrm{d}(\nabla\varphi_1^*)_\#\mu
$$

$$
\leq \int_\Omega \mathcal{B}_{\varphi_2^*}(\nabla\varphi_2 \circ \nabla\varphi_1^*(x), x)\,\mathrm{d}\mu \leq
$$

$$
\frac{1}{2\alpha}\int_\Omega \|\nabla\varphi_2(x) - \nabla\varphi_1(x)\|_2^2\,\mathrm{d}(\nabla\varphi_1^*)_\#\mu,
$$

where we used the fact that $\nabla\varphi_1 \circ \nabla\varphi_1^* = \mathrm{id}$. The density of $(\nabla\varphi_1^*)_\#\mu$ is $\rho = \mu \circ \nabla\varphi_1 \cdot |D_x\nabla\varphi_1|$. Since $\varphi_1$ is $\alpha$-strongly convex and $\beta$-smooth, we have $a\alpha^d \leq \rho(x) \leq b\beta^d$ and thus

$$
\frac{a\alpha^d}{2\beta}\|\varphi_2 - \varphi_1\|_{\dot{\mathbb{H}}^1}^2 \leq \int_\Omega \mathcal{B}_{\varphi_2^*}(\nabla\varphi_2 \circ \nabla\varphi_1^*(x), x)\,\mathrm{d}\mu \leq \frac{b\beta^d}{2\alpha}\|\varphi_2 - \varphi_1\|_{\dot{\mathbb{H}}^1}^2.
$$

$\square$

## B.2 PROOF OF LEMMA 2

*Proof.* We first show the uniqueness of maximizer. Suppose there exits two maximizers $\varphi_1 \neq \varphi_2$, then for any $\gamma \in [0, 1]$, $\varphi_t := \gamma\varphi_1 + (1 - \gamma)\varphi_2$ is again a maximizer. Applying Lemma 1,

$$-\frac{A(1-\gamma)^2}{2}\|\varphi_2 - \varphi_1\|_{\dot{\mathbb{H}}^1}^2 \geq \mathcal{I}_\nu^\mu(\varphi_t) - \mathcal{I}_\nu^\mu(\varphi_1) - (1 - \gamma)\langle\boldsymbol{\nabla}\mathcal{I}_\nu^\mu(\varphi_t), \varphi_2 - \varphi_1\rangle_{\dot{\mathbb{H}}^1} \tag{8}$$

$$-\frac{A\gamma^2}{2}\|\varphi_2 - \varphi_1\|_{\dot{\mathbb{H}}^1}^2 \geq \mathcal{I}_\nu^\mu(\varphi_t) - \mathcal{I}_\nu^\mu(\varphi_2) - \gamma\langle\boldsymbol{\nabla}\mathcal{I}_\nu^\mu(\varphi_t), \varphi_1 - \varphi_2\rangle_{\dot{\mathbb{H}}^1} \tag{9}$$

Adding equation 8 multiplied by $\gamma$ and equation 9 multiplied by $1 - \gamma$ gives

$$-\frac{A(1-\gamma)\gamma}{2}\|\varphi_2 - \varphi_1\|_{\dot{\mathbb{H}}^1}^2 \geq \mathcal{I}_\nu^\mu(\varphi_t) - \gamma\mathcal{I}_\nu^\mu(\varphi_1) - (1 - \gamma)\mathcal{I}_\nu^\mu(\varphi_2).$$

For fixed $\gamma \in (0, 1)$, the right-hand side of above is 0, while the left-hand side is strictly smaller than 0. This shows a contradiction and the uniqueness is proved.

Next, we show that the first variation is given by $\frac{\delta\mathcal{L}^\mu}{\delta\nu}(\nu) = \int_\Omega \frac{\|\mathrm{id}\|_2^2}{2} - \widetilde{\varphi}_\nu^\mu \, \mathrm{d}\chi$. Note that

$$\frac{\mathrm{d}}{\mathrm{d}\epsilon}\mathcal{L}^\mu(\nu + \epsilon\chi)\bigg|_{\epsilon=0} = \frac{\max_{\varphi\in\mathbb{F}_{\alpha,\beta}}\mathcal{I}_{\nu+\epsilon\chi}^\mu(\varphi) - \max_{\varphi\in\mathbb{F}_{\alpha,\beta}}\mathcal{I}_\nu^\mu(\varphi)}{\epsilon}$$

$$\leq \frac{\mathcal{I}_{\nu+\epsilon\chi}^\mu(\widetilde{\varphi}_{\nu+\epsilon\chi}^\mu) - \mathcal{I}_\nu^\mu(\widetilde{\varphi}_{\nu+\epsilon\chi}^\mu)}{\epsilon}$$

$$= \int_\Omega \frac{\langle\mathrm{id}, \mathrm{id}\rangle}{2} - \widetilde{\varphi}_{\nu+\epsilon\chi}^\mu \, \mathrm{d}\chi$$

$$\to \int_\Omega \frac{\langle\mathrm{id}, \mathrm{id}\rangle}{2} - \widetilde{\varphi}_\nu^\mu \, \mathrm{d}\chi \qquad \text{as } \epsilon \to 0.$$

On the other hand, we have

$$\frac{\max_{\varphi\in\mathbb{F}_{\alpha,\beta}}\mathcal{I}_{\nu+\epsilon\chi}^\mu(\varphi) - \max_{\varphi\in\mathbb{F}_{\alpha,\beta}}\mathcal{I}_\nu^\mu(\varphi)}{\epsilon} \geq \frac{\mathcal{I}_{\nu+\epsilon\chi}^\mu(\widetilde{\varphi}_\nu^\mu) - \mathcal{I}_\nu^\mu(\widetilde{\varphi}_\nu^\mu)}{\epsilon} = \int_\Omega \frac{\langle\mathrm{id}, \mathrm{id}\rangle}{2} - \widetilde{\varphi}_\nu^\mu \, \mathrm{d}\chi.$$

Recall that the Wasserstein gradient is just the standard gradient of the first variation. Together, the Lemma is concluded.

## B.3 PROOF OF LEMMA 3

The equality part is direct by applying Lemma 2,

$$\|\mathbb{W}\mathcal{L}^\mu(\nu_1) - \mathbb{W}\mathcal{L}^\mu(\nu_2)\|_{L^2} = \|\nabla\widetilde{\varphi}_{\nu_1}^\mu - \nabla\widetilde{\varphi}_{\nu_2}^\mu\|_{L^2} = \|\varphi_{\nu_1}^\mu - \varphi_{\nu_2}^\mu\|_{\dot{\mathbb{H}}^1}.$$

To prove the inequality part, note that $\mathcal{I}_{\nu,\mu}$ is concave by Lemma 1, and $\mathbb{F}_{\alpha,\beta}$ is a convex set. By the optimality of $\widetilde{\varphi}_{\nu_2}^\mu, \widetilde{\varphi}_{\nu_1}^\mu$, for any $\varphi \in \mathbb{F}_{\alpha,\beta}$, we have

$$\langle\varphi - \widetilde{\varphi}_{\nu_2}^\mu, \boldsymbol{\nabla}\mathcal{I}_{\nu_2}^\mu(\widetilde{\varphi}_{\nu_2}^\mu)\rangle_{\dot{\mathbb{H}}^1} \leq 0, \tag{10}$$

$$\langle\varphi - \widetilde{\varphi}_{\nu_1}^\mu, \boldsymbol{\nabla}\mathcal{I}_{\nu_1}^\mu(\widetilde{\varphi}_{\nu_1}^\mu)\rangle_{\dot{\mathbb{H}}^1} \leq 0. \tag{11}$$

Substituting $\varphi = \widetilde{\varphi}_{\nu_1}^\mu$ in equation 10 and $\varphi = \widetilde{\varphi}_{\nu_2}^\mu$ in equation 11 and summing them together results

$$\langle\widetilde{\varphi}_{\nu_1}^\mu - \widetilde{\varphi}_{\nu_2}^\mu, \boldsymbol{\nabla}\mathcal{I}_{\nu_2}^\mu(\widetilde{\varphi}_{\nu_2}^\mu) - \boldsymbol{\nabla}\mathcal{I}_{\nu_1}^\mu(\widetilde{\varphi}_{\nu_1}^\mu)\rangle_{\dot{\mathbb{H}}^1} \leq 0. \tag{12}$$

The following two inequalities follow from Lemma 1,

$$\mathcal{I}_{\nu_1}^\mu(\widetilde{\varphi}_{\nu_1}^\mu) - \mathcal{I}_{\nu_1}^\mu(\widetilde{\varphi}_{\nu_2}^\mu) - \langle\boldsymbol{\nabla}\mathcal{I}_{\nu_1}^\mu(\widetilde{\varphi}_{\nu_2}^\mu), \widetilde{\varphi}_{\nu_1}^\mu - \widetilde{\varphi}_{\nu_2}^\mu\rangle_{\dot{\mathbb{H}}^1} \leq -\frac{A}{2}\|\widetilde{\varphi}_{\nu_2}^\mu - \widetilde{\varphi}_{\nu_1}^\mu\|_{\dot{\mathbb{H}}^1}^2,$$

$$\mathcal{I}_{\nu_1}^\mu(\widetilde{\varphi}_{\nu_2}^\mu) - \mathcal{I}_{\nu_1}^\mu(\widetilde{\varphi}_{\nu_1}^\mu) - \langle\boldsymbol{\nabla}\mathcal{I}_{\nu_1}^\mu(\widetilde{\varphi}_{\nu_1}^\mu), \widetilde{\varphi}_{\nu_2}^\mu - \widetilde{\varphi}_{\nu_1}^\mu\rangle_{\dot{\mathbb{H}}^1} \leq -\frac{A}{2}\|\widetilde{\varphi}_{\nu_2}^\mu - \widetilde{\varphi}_{\nu_1}^\mu\|_{\dot{\mathbb{H}}^1}^2.$$

Summing over the above two inequalities gives

$$\langle\boldsymbol{\nabla}\mathcal{I}_{\nu_1}^\mu(\widetilde{\varphi}_{\nu_1}^\mu) - \boldsymbol{\nabla}\mathcal{I}_{\nu_1}^\mu(\widetilde{\varphi}_{\nu_2}^\mu), \widetilde{\varphi}_{\nu_1}^\mu - \widetilde{\varphi}_{\nu_2}^\mu\rangle_{\dot{\mathbb{H}}^1} \leq -A\|\widetilde{\varphi}_{\nu_2}^\mu - \widetilde{\varphi}_{\nu_1}^\mu\|_{\dot{\mathbb{H}}^1}^2.$$

Then, combining the above inequality with equation 12 shows that

$$
\begin{aligned}
A\|\widetilde{\varphi}^\mu_{\nu_2} - \widetilde{\varphi}^\mu_{\nu_1}\|^2_{\dot{\mathbb{H}}^1} \leq & \langle \boldsymbol{\nabla}\mathcal{I}^\mu_{\nu_1}(\widetilde{\varphi}^\mu_{\nu_2}) - \boldsymbol{\nabla}\mathcal{I}^\mu_{\nu_2}(\widetilde{\varphi}^\mu_{\nu_2}), \widetilde{\varphi}^\mu_{\nu_1} - \widetilde{\varphi}^\mu_{\nu_2} \rangle_{\dot{\mathbb{H}}^1} \\
& \overset{\text{(i)}}{=} \int_\Omega \widetilde{\varphi}^\mu_{\nu_2} - \widetilde{\varphi}^\mu_{\nu_1} \, \mathrm{d}(\nu_1 - \nu_2) \\
& \leq \|\widetilde{\varphi}^\mu_{\nu_1} - \widetilde{\varphi}^\mu_{\nu_2}\|_{\dot{\mathbb{H}}^1} \|\nu_1 - \nu_2\|_{\dot{\mathbb{H}}^{-1}}.
\end{aligned}
$$

Here, (i) is derived by applying Lemma 5 as

$$
\begin{aligned}
& \langle \boldsymbol{\nabla}\mathcal{I}^\mu_{\nu_1}(\widetilde{\varphi}^\mu_{\nu_2}), \widetilde{\varphi}^\mu_{\nu_1} - \widetilde{\varphi}^\mu_{\nu_2} \rangle - \langle \boldsymbol{\nabla}\mathcal{I}^\mu_{\nu_2}(\widetilde{\varphi}^\mu_{\nu_2}), \widetilde{\varphi}^\mu_{\nu_1} - \widetilde{\varphi}^\mu_{\nu_2} \rangle \\
& = \Big[ \int_\Omega \widetilde{\varphi}^\mu_{\nu_2} - \widetilde{\varphi}^\mu_{\nu_1} \, \mathrm{d}\nu_1 - \int_\Omega \big[ \widetilde{\varphi}^\mu_{\nu_2} \circ \nabla(\widetilde{\varphi}^\mu_{\nu_2})^* - \widetilde{\varphi}^\mu_{\nu_1} \circ \nabla(\widetilde{\varphi}^\mu_{\nu_2})^* \big] \, \mathrm{d}\mu \Big] \\
& \quad - \Big[ \int_\Omega \widetilde{\varphi}^\mu_{\nu_2} - \widetilde{\varphi}^\mu_{\nu_1} \, \mathrm{d}\nu_2 - \int_\Omega \big[ \widetilde{\varphi}^\mu_{\nu_2} \circ \nabla(\widetilde{\varphi}^\mu_{\nu_2})^* - \widetilde{\varphi}^\mu_{\nu_1} \circ \nabla(\widetilde{\varphi}^\mu_{\nu_2})^* \big] \, \mathrm{d}\mu \Big] \\
& = \int_\Omega \widetilde{\varphi}^\mu_{\nu_2} - \widetilde{\varphi}^\mu_{\nu_1} \, \mathrm{d}(\nu_1 - \nu_2).
\end{aligned}
$$

$\square$

## B.4 PROOF OF LEMMA 4

*Proof.* Notice that

$$
\begin{aligned}
& \mathcal{L}^\mu(\nu_2) - \mathcal{L}^\mu(\nu_1) - \int_\Omega \frac{\langle \mathrm{id}, \mathrm{id} \rangle}{2} - \widetilde{\varphi}^\mu_{\nu_1} \, \mathrm{d}\nu_2 - \nu_1 \\
& = \int_0^1 \int_\Omega \frac{\langle \mathrm{id}, \mathrm{id} \rangle}{2} - \widetilde{\varphi}^\mu_{\nu_1+\epsilon(\nu_2-\nu_1)} \, \mathrm{d}\nu_2 - \nu_1 \, \mathrm{d}\epsilon - \int_0^1 \int_\Omega \frac{\langle \mathrm{id}, \mathrm{id} \rangle}{2} - \widetilde{\varphi}^\mu_{\nu_1} \, \mathrm{d}\nu_2 - \nu_1 \, \mathrm{d}\epsilon \\
& = \int_0^1 \int_\Omega \widetilde{\varphi}^\mu_{\nu_1} - \widetilde{\varphi}^\mu_{\nu_1+\epsilon(\nu_2-\nu_1)} \, \mathrm{d}\nu_2 - \nu_1 \, \mathrm{d}\epsilon \\
& \leq \int_0^1 \|\widetilde{\varphi}^\mu_{\nu_1} - \widetilde{\varphi}^\mu_{\nu_1+\epsilon(\nu_2-\nu_1)}\|_{\dot{\mathbb{H}}^1} \|\nu_2 - \nu_1\|_{\dot{\mathbb{H}}^{-1}} \, \mathrm{d}\epsilon \\
& \overset{\text{(i)}}{\leq} \int_0^1 \frac{1}{A} \|\varepsilon(\nu_2 - \nu_1)\|_{\dot{\mathbb{H}}^{-1}} \cdot \|\nu_2 - \nu_1\|_{\dot{\mathbb{H}}^{-1}} \, \mathrm{d}\varepsilon = \frac{1}{2A} \|\nu_1 - \nu_2\|^2_{\dot{\mathbb{H}}^{-1}} \\
& \overset{\text{(ii)}}{\leq} \frac{\max\{\|\nu_1\|_\infty, \|\nu_2\|_\infty\}}{2A} \mathcal{W}^2_2(\nu_1, \nu_2),
\end{aligned}
$$

where (i) is due to Lemma 3, and (ii) follows from Theorem 5.34 (Santambrogio, 2015). Since $\frac{\langle \mathrm{id}, \mathrm{id} \rangle}{2} - \widetilde{\varphi}^\mu_{\nu_1}$ is $(1+\alpha)$-smooth, we have

$$
\begin{aligned}
& \int_\Omega \frac{\langle \mathrm{id}, \mathrm{id} \rangle}{2} - \widetilde{\varphi}^\mu_{\nu_1} \, \mathrm{d}\nu_2 - \nu_1 \\
& = \int_\Omega (\frac{\langle \mathrm{id}, \mathrm{id} \rangle}{2} - \widetilde{\varphi}^\mu_{\nu_1}) \circ T^{\nu_2}_{\nu_1} - (\frac{\langle \mathrm{id}, \mathrm{id} \rangle}{2} - \widetilde{\varphi}^\mu_{\nu_1}) \circ \mathrm{id} \, \mathrm{d}\nu_1 \\
& \leq \int_\Omega \langle \mathrm{id} - \nabla\widetilde{\varphi}^\mu_{\nu_1}, T^{\nu_2}_{\nu_1} - \mathrm{id} \rangle + \frac{1+\alpha}{2} \|T^{\nu_2}_{\nu_1} - \mathrm{id}\|^2_2 \, \mathrm{d}\nu_1.
\end{aligned}
$$

Combining the above two results would conclude the lemma. $\square$

## B.5 PROOF OF THEOREM 1

**Lemma 6.** *For any $\varphi^t \in \mathbb{F}_{\alpha,\beta}(\Omega)$, let $\varphi^{t+1} = \mathcal{P}_{\mathbb{F}_{\alpha,\beta}}(\varphi^t + \eta\boldsymbol{\nabla}\mathcal{I}^\mu_\nu(\varphi^t))$. If $\eta \leq 1/B$, then*

$$
\|\varphi^{t+1} - \widetilde{\varphi}^\mu_\nu\|^2_{\dot{\mathbb{H}}^1} \leq (1 - A\eta)\|\varphi^t - \widetilde{\varphi}^\mu_\nu\|^2_{\dot{\mathbb{H}}^1}.
$$

*Proof.* Recall that $\widetilde{\varphi}_\nu^\mu = \arg\max_{\varphi \in \mathbb{F}_{\alpha,\beta}} \mathcal{I}_\nu^\mu(\varphi)$. We have

$$\|\varphi^{t+1} - \widetilde{\varphi}_\nu^\mu\|_{\dot{\mathbb{H}}^1}^2$$

$$\overset{(i)}{\le} \|\varphi^t + \eta\boldsymbol{\nabla}\mathcal{I}_\nu^\mu(\varphi^t) - \widetilde{\varphi}_\nu^\mu\|_{\dot{\mathbb{H}}^1}^2$$

$$= \|\varphi^t - \widetilde{\varphi}_\nu^\mu\|_{\dot{\mathbb{H}}^1}^2 + \eta^2\|\boldsymbol{\nabla}\mathcal{I}_\nu^\mu(\varphi^t)\|_{\dot{\mathbb{H}}^1}^2 + 2\eta\langle\boldsymbol{\nabla}\mathcal{I}_\nu^\mu(\varphi^t), \varphi^t - \widetilde{\varphi}_\nu^\mu\rangle_{\dot{\mathbb{H}}^1}$$

$$\overset{(ii)}{\le} \|\varphi^t - \widetilde{\varphi}_\nu^\mu\|_{\dot{\mathbb{H}}^1}^2 + \eta^2\|\boldsymbol{\nabla}\mathcal{I}_\nu^\mu(\varphi^t)\|_{\dot{\mathbb{H}}^1}^2 + 2\eta\left(\mathcal{I}_\nu^\mu(\varphi^t) - \mathcal{I}_\nu^\mu(\widetilde{\varphi}_\nu^\mu) - \frac{A}{2}\|\varphi^t - \widetilde{\varphi}_\nu^\mu\|_{\dot{\mathbb{H}}^1}^2\right).$$

Here, (i) is due to the property of the projection map, and (ii) is by Lemma 1. Since $\dot{\mathbb{H}}^1$ is a linear space and $\varphi^t + \boldsymbol{\nabla}\mathcal{I}_\nu^\mu(\varphi^t)/B \in \dot{\mathbb{H}}^1$, we have

$$\mathcal{I}_\nu^\mu(\widetilde{\varphi}_\nu^\mu) \ge \mathcal{I}_\nu^\mu(\varphi^t + \frac{1}{B}\boldsymbol{\nabla}\mathcal{I}_\nu^\mu(\varphi^t))$$

$$\overset{(i)}{\ge} \mathcal{I}_\nu^\mu(\varphi^t) + \langle\boldsymbol{\nabla}\mathcal{I}_\nu^\mu(\varphi^t), \frac{1}{B}\boldsymbol{\nabla}\mathcal{I}_\nu^\mu(\varphi^t)\rangle_{\dot{\mathbb{H}}^1} - \frac{B}{2}\|\frac{1}{B}\boldsymbol{\nabla}\mathcal{I}_\nu^\mu(\varphi^t)\|_{\dot{\mathbb{H}}^1}^2$$

$$= \mathcal{I}_\nu^\mu(\varphi^t) + \frac{1}{2B}\|\boldsymbol{\nabla}\mathcal{I}(\varphi^t)\|_{\dot{\mathbb{H}}^1}^2.$$

Again, (i) is due to Lemma 1. Combining the above two inequalities yields

$$\|\varphi^{t+1} - \widetilde{\varphi}_\nu^\mu\|_{\dot{\mathbb{H}}^1}^2$$

$$\le \|\varphi^t - \widetilde{\varphi}_\nu^\mu\|_{\dot{\mathbb{H}}^1}^2 + 2B\eta^2(\mathcal{I}_\nu^\mu(\widetilde{\varphi}_\nu^\mu) - \mathcal{I}_\nu^\mu(\varphi^t)) + 2\eta\left(\mathcal{I}_\nu^\mu(\varphi^t) - \mathcal{I}_\nu^\mu(\widetilde{\varphi}_\nu^\mu) - \frac{A}{2}\|\varphi^t - \widetilde{\varphi}_\nu^\mu\|_{\dot{\mathbb{H}}^1}^2\right)$$

$$= (1 - A\eta)\|\varphi^t - \widetilde{\varphi}_\nu^\mu\|_{\dot{\mathbb{H}}^1}^2 + 2\eta(1 - B\eta)\left(\mathcal{I}_\nu^\mu(\varphi^t) - \mathcal{I}_\nu^\mu(\widetilde{\varphi}_\nu^\mu)\right).$$

If $\eta \le 1/B$, we have $\|\varphi^{t+1} - \widetilde{\varphi}_\nu^\mu\|_{\dot{\mathbb{H}}^1}^2 \le (1 - A\eta)\|\varphi^t - \widetilde{\varphi}_\nu^\mu\|_{\dot{\mathbb{H}}^1}^2$. $\qquad\square$

*Proof of Theorem 1.* Since $\nu^{t+1} = (\mathrm{id} - \tau(\mathrm{id} - \nabla\overline{\varphi}^t))_\#\nu^t$, where $\overline{\varphi}^t = \frac{1}{n}\sum_{i=1}^n \varphi_{\nu^t}^{\mu_i}$, we have from Lemma 4 that for each $i$,

$$\mathcal{L}^{\mu_i}(\nu^{t+1}) - \mathcal{L}^{\mu_i}(\nu^t) \le \tau\int_\Omega \langle\mathrm{id} - \nabla\widetilde{\varphi}_{\nu^t}^{\mu_i}, \nabla\overline{\varphi}^t - \mathrm{id}\rangle\,\mathrm{d}\nu^t + \tau^2\frac{C_1}{2}\int_\Omega \|\nabla\overline{\varphi}^t - \mathrm{id}\|_2^2\,\mathrm{d}\nu^t,$$

where $C_1 = 1 + \alpha + \frac{V}{A}$. Averaging over $i$ yields

$$\mathcal{F}_{\alpha,\beta}(\nu^{t+1}) - \mathcal{F}_{\alpha,\beta}(\nu^t)$$

$$\le \tau\int_\Omega \langle\mathrm{id} - \nabla\overline{\widetilde{\varphi}}^t, \nabla\overline{\varphi}^t - \mathrm{id}\rangle\,\mathrm{d}\nu^t + \tau^2\frac{C_1}{2}\int_\Omega \|\nabla\overline{\varphi}^t - \mathrm{id}\|_2^2\,\mathrm{d}\nu^t$$

$$= \frac{\tau}{2}\int_\Omega \|\nabla\overline{\widetilde{\varphi}}^t - \nabla\overline{\varphi}^t\|_2^2\,\mathrm{d}\nu^t - \frac{\tau}{2}\int_\Omega \|\nabla\overline{\widetilde{\varphi}}^t - \mathrm{id}\|_2^2\,\mathrm{d}\nu^t - \frac{\tau - \tau^2 C_1}{2}\int_\Omega \|\nabla\overline{\varphi}^t - \mathrm{id}\|_2^2\,\mathrm{d}\nu^t$$

$$\overset{(i)}{\le} \frac{2\tau - \tau^2 C_1}{2}\int_\Omega \|\nabla\overline{\widetilde{\varphi}}^t - \nabla\overline{\varphi}^t\|_2^2\,\mathrm{d}\nu^t - \frac{3\tau - \tau^2 C_1}{4}\int_\Omega \|\nabla\overline{\widetilde{\varphi}}^t - \mathrm{id}\|_2^2\,\mathrm{d}\nu^t$$

$$\le \frac{(2\tau - \tau^2 C_1)V}{2}\int_\Omega \|\nabla\overline{\widetilde{\varphi}}^t - \nabla\overline{\varphi}^t\|_2^2\,\mathrm{d}x - \frac{3\tau - \tau^2 C_1}{4}\int_\Omega \|\nabla\overline{\widetilde{\varphi}}^t - \mathrm{id}\|_2^2\,\mathrm{d}\nu^t$$

$$\le \tau V\int_\Omega \|\nabla\overline{\widetilde{\varphi}}^t - \nabla\overline{\varphi}^t\|_2^2\,\mathrm{d}x - \frac{3\tau - \tau^2 C_1}{4}\int_\Omega \|\nabla\overline{\widetilde{\varphi}}^t - \mathrm{id}\|_2^2\,\mathrm{d}\nu^t$$

$$\le \frac{\tau V}{n}\sum_{i=1}^n \int_\Omega \|\nabla\widetilde{\varphi}_{\nu^t}^{\mu_i} - \nabla\varphi_i^t\|_2^2\,\mathrm{d}x - \frac{3\tau - \tau^2 C_1}{4}\int_\Omega \|\nabla\overline{\widetilde{\varphi}}^t - \mathrm{id}\|_2^2\,\mathrm{d}\nu^t \qquad (13)$$

where (i) is due to the fact $\frac{1}{2}\int_\Omega \|\nabla\overline{\widetilde{\varphi}}^t - \mathrm{id}\|_2^2\,\mathrm{d}\nu^t \le \int_\Omega \|\nabla\overline{\widetilde{\varphi}}^t - \nabla\overline{\varphi}^t\|_2^2\,\mathrm{d}\nu^t + \int_\Omega \|\nabla\overline{\varphi}^t - \mathrm{id}\|_2^2\,\mathrm{d}\nu^t$. Set $\delta_i^t = \int_\Omega \|\nabla\widetilde{\varphi}_{\nu^t}^{\mu_i} - \nabla\varphi_i^t\|_2^2\,\mathrm{d}x$. By Young's inequality,

$$\delta_i^t \le \left[1 + \frac{1}{2(\frac{1}{A\eta} - 1)}\right]\int_\Omega \|\nabla\widetilde{\varphi}_{\nu^{t-1}}^{\mu_i} - \nabla\varphi_i^t\|_2^2\,\mathrm{d}x + \left[1 + 2\left(\frac{1}{A\eta} - 1\right)\right]\int_\Omega \|\nabla\widetilde{\varphi}_{\nu^t}^{\mu_i} - \nabla\widetilde{\varphi}_{\nu^{t-1}}^{\mu_i}\|_2^2\,\mathrm{d}x.$$

For the first term above, applying Lemma 6 yields

$$\left[1 + \frac{1}{2(\frac{1}{A\eta}-1)}\right] \int_\Omega \|\nabla\widetilde{\varphi}^{\mu_i}_{\nu^{t-1}} - \nabla\varphi^t_i\|^2_2 \,\mathrm{d}x \le \left(1 - \frac{A\eta}{2}\right)\delta^{t-1}_i.$$

For the second term, Lemma 2 and Theorem 5.34 (Santambrogio, 2015) implies that

$$\int_\Omega \|\nabla\widetilde{\varphi}^{\mu_i}_{\nu^{t-1}} - \nabla\widetilde{\varphi}^{\mu_i}_{\nu^t}\|^2_2 \,\mathrm{d}x = \|\widetilde{\varphi}^{\mu_i}_{\nu^{t-1}} - \widetilde{\varphi}^{\mu_i}_{\nu^t}\|^2_{\dot{\mathbb{H}}^1}$$

$$\overset{(i)}{\le} \frac{1}{A^2}\|\nu^{t-1} - \nu^t\|^2_{\dot{\mathbb{H}}^{-1}} \overset{(ii)}{\le} \frac{V}{A^2}\mathcal{W}^2_2(\nu^{t-1}, \nu^t)$$

$$\overset{(iii)}{\le} \frac{V}{A^2}\int_\Omega \|(\mathrm{id} - \tau\mathbb{W}\mathcal{J}(\nu^{t-1}, \boldsymbol{\varphi}^{t-1})) - \mathrm{id}\|^2_2 \,\mathrm{d}\nu^{t-1} = \frac{\tau^2 V}{A^2}\int_\Omega \|\mathrm{id} - \nabla\overline{\varphi}^{t-1}\|^2_2 \,\mathrm{d}\nu^{t-1}$$

$$\le \frac{2\tau^2 V}{A^2}\left(\int_\Omega \|\nabla\overline{\widetilde{\varphi}}^{t-1} - \nabla\overline{\varphi}^{t-1}\|^2_2 \,\mathrm{d}\nu^{t-1} + \int_\Omega \|\nabla\overline{\widetilde{\varphi}}^{t-1} - \mathrm{id}\|^2_2 \,\mathrm{d}\nu^{t-1}\right)$$

$$\le \frac{2\tau^2 V}{A^2}\left(\frac{V}{n}\sum_{i=1}^n \delta^{t-1}_i + \int_\Omega \|\nabla\overline{\widetilde{\varphi}}^{t-1} - \mathrm{id}\|^2_2 \,\mathrm{d}\nu^{t-1}\right).$$

Here, (i) is due to Lemma 3, (ii) is due to equation 6 (Theorem 5.34, Santambrogio, 2015), and (iii) is because $\mathrm{id} - \tau\mathbb{W}\mathcal{J}(\nu^{t-1}, \boldsymbol{\varphi}^{t-1})$ is a transport map from $\nu^{t-1}$ to $\nu^t$. Combining above pieces together yields

$$\delta^t_i \le \left(1 - \frac{A\eta}{2}\right)\delta^{t-1}_i + \left[1 + 2\left(\frac{1}{A\eta}-1\right)\right]\frac{2\tau^2 V}{A^2}\left(\frac{V}{n}\sum_{i=1}^n \delta^{t-1}_i + \int_\Omega \|\nabla\overline{\widetilde{\varphi}}^{t-1} - \mathrm{id}\|^2_2 \,\mathrm{d}\nu^{t-1}\right).$$

Set $\bar{\delta}^t = \frac{1}{n}\sum_{i=1}^n \delta^t_i$ and $\gamma = 1 - \frac{A\eta}{2} + \frac{2\tau^2 V^2(2-A\eta)}{A^3\eta}$. Averaging the above inequality for $i \in \{1, \cdots, n\}$ yields

$$\bar{\delta}^t \le \gamma\bar{\delta}^{t-1} + \frac{2\tau^2 V(2 - A\eta)}{A^3\eta}\int_\Omega \|\nabla\overline{\widetilde{\varphi}}^{t-1} - \mathrm{id}\|^2_2 \,\mathrm{d}\nu^{t-1}$$

$$\le \gamma^{t-1}\bar{\delta}^1 + \frac{2\tau^2 V(2 - A\eta)}{A^3\eta}\sum_{k=1}^{t-1}\gamma^{t-k-1}\int_\Omega \|\nabla\overline{\widetilde{\varphi}}^k - \mathrm{id}\|^2_2 \,\mathrm{d}\nu^k. \tag{14}$$

Putting all pieces together yields

$$\mathcal{F}_{\alpha,\beta}(\nu^{T+1}) - \mathcal{F}_{\alpha,\beta}(\nu^1) = \sum_{t=1}^T \left[\mathcal{F}_{\alpha,\beta}(\nu^{t+1}) - \mathcal{F}_{\alpha,\beta}(\nu^t)\right]$$

$$\overset{(i)}{\le} \sum_{t=1}^T \left[\tau V\bar{\delta}^t - \frac{3\tau - \tau^2 C_1}{4}\int_\Omega \|\nabla\overline{\widetilde{\varphi}}^t - \mathrm{id}\|^2_2 \,\mathrm{d}\nu^t\right]$$

$$\overset{(ii)}{\le} \sum_{t=1}^T \left[\tau V\gamma^{t-1}\bar{\delta}^1 + \frac{2\tau^3 V^2(2 - A\eta)}{A^3\eta}\sum_{k=1}^{t-1}\gamma^{t-k-1}\int_\Omega \|\nabla\overline{\widetilde{\varphi}}^k - \mathrm{id}\|^2_2 \,\mathrm{d}\nu^k\right.$$

$$\left. - \frac{3\tau - \tau^2 C_1}{4}\int_\Omega \|\nabla\overline{\widetilde{\varphi}}^t - \mathrm{id}\|^2_2 \,\mathrm{d}\nu^t\right]$$

$$= \tau V\bar{\delta}^1 \cdot \frac{1 - \gamma^T}{1 - \gamma} + \sum_{t=1}^T \left[\frac{2\tau^3 V^2(2 - A\eta)}{A^3\eta} \cdot \frac{1 - \gamma^{T-t}}{1 - \gamma} - \frac{3\tau - \tau^2 C_1}{4}\right]\int_\Omega \|\nabla\overline{\widetilde{\varphi}}^t - \mathrm{id}\|^2_2 \,\mathrm{d}\nu^t$$

$$\overset{(iii)}{\le} \frac{4\tau V\bar{\delta}^1}{A\eta} + \left[\frac{2\tau^3 V^2(2 - A\eta)}{A^3\eta} \cdot \frac{4}{A\eta} - \frac{3\tau - \tau^2 C_1}{4}\right]\sum_{t=1}^T \int_\Omega \|\nabla\overline{\widetilde{\varphi}}^t - \mathrm{id}\|^2_2 \,\mathrm{d}\nu^t$$

$$= \frac{4\tau V\bar{\delta}^1}{A\eta} + \left[\frac{8\tau^3 V^2(2 - A\eta)}{A^4\eta^2} - \frac{3\tau - \tau^2 C_1}{4}\right]\sum_{t=1}^T \int_\Omega \|\nabla\overline{\widetilde{\varphi}}^t - \mathrm{id}\|^2_2 \,\mathrm{d}\nu^t.$$

| | reg | Wasserstein distance | $L^2$-distance | $\mathcal{F}(\nu^{est})$ |
|---|---|---|---|---|
| WDHA | | $3.758 \times 10^{-8}$ | 0.4869 | $9.001 \times 10^{-2}$ |
| CWB | 0.003 | $2.98 \times 10^{-4}$ | 1.321 | $9.031 \times 10^{-2}$ |
| | 0.002 | $2.538 \times 10^{-2}$ | 3.041 | 0.1154 |
| DSB | 0.005 | $1.2 \times 10^{-4}$ | 0.8219 | $9.013 \times 10^{-2}$ |
| | 0.004 | $1.164 \times 10^{-2}$ | 2.281 | 0.1016 |

Table 1: Simulation results for uniform distributions supported on round disks.

Here, (i) is from equation 13, (ii) is due to equation 14, and in (iii) we use the fact that

$$1 - \gamma = \frac{A\eta}{2} - \frac{2\tau^2 V^2 (2 - A\eta)}{A^3 \eta} > \frac{A\eta}{4}$$

when $\tau < \frac{A^2 \eta}{2V\sqrt{2(2-A\eta)}}$. Therefore, we have

$$\frac{1}{T} \sum_{t=1}^{T} \int_{\Omega} \|\nabla \overline{\varphi}^t - \mathrm{id}\|_2^2 \, d\nu^t \leq \frac{1}{T} \cdot \frac{\frac{4\tau V \bar{\delta}^1}{A\eta} + \mathcal{F}_{\alpha,\beta}(\nu^1) - \mathcal{F}_{\alpha,\beta}(\nu^{T+1})}{\frac{3}{4}\tau - \frac{C_1}{4}\tau^2 - \frac{8V^2(2-A\eta)}{A^4\eta^2}\tau^3}$$

$$< \frac{\frac{4\tau V \bar{\delta}^1}{A\eta} + \mathcal{F}_{\alpha,\beta}(\nu^1) - \mathcal{F}_{\alpha,\beta}(\nu^{T+1})}{T\tau/2},$$

where the last inequality holds when $\tau < \frac{1}{C_1 + 2\sqrt{\frac{8V^2(2-A\eta)}{A^4\eta^2}}} = \frac{A^2\eta}{A\eta(A\alpha + A + V) + 4V\sqrt{4 - 2A\eta}}.$ □

## C  ADDITIONAL EMPIRICAL STUDIES

### C.1  UNIFORM DISTRIBUTIONS WITH GROUND TRUTH

Here, the goal is to compute the barycenter of four uniform distributions supported on round disks of radius 0.15, centered at $(0.2, 0.2)$, $(0.2, 0.8)$, $(0.8, 0.2)$, $(0.8, 0.8)$ respectively. It's clear that the true barycenter is uniform on the disk of radius 0.15 centered at $(0.5, 0.5)$. The computed barycenter densities by WDHA, CWB and DSB are shown in Figure 3. We note that regularization parameters reg= 0.003 and reg=0.005 are the optimal choices for CWB and DSB respectively. Smaller regularization parameters lead nonconvergent and worse results for both CWB and DSB. We report in Table 1 the Wasserstein distance between computed barycenter distribution and the truth, $L^2$-distance between computed barycenter densities and the true density, and the barycenter functional value. Our method is uniformly the best, and in particular, the improvement in the Wasserstein distance is of orders of magnitude.

### C.2  EXPERIMENTS ON 1D DISTRIBUTIONS

In this empirical study, we compare the performance of Algorithm 3 (with projection onto $\mathbb{F}_{\alpha,\beta}$) and Algorithm 4 (with double convex conjugates) on 1D distributions. For each repetition $t$ and $i = 1, 2, 3$, we let $\mu_{i,t}$ be the truncated version of $N(a_i, \sigma_i^2)$ on the domain $[0, 1]$, where $a_i \sim$ uniform $[0.3, 0.7]$ and $\sigma_i \sim$ uniform $[0, 1]$. Let $\bar{\nu}_t, \bar{\nu}_{1,t}, \bar{\nu}_{2,t}$ be the true barycenter, computed barycenter from Algorithm 3, computed barycenter from Algorithm 4 respectively. We repeat the experiment 300 times and report two types of average distances between the groundtruth and each computed barycenters : average $\mathcal{W}_2$-distance $\frac{1}{T} \sum_{t=1}^{T} W_2(\bar{\nu}_t, \bar{\nu}_{j,t})$ and average $L^2$-distance between densities $\frac{1}{T} \sum_{t=1}^{T} (\int (\bar{\nu}_t(x) - \bar{\nu}_{j,t}(x))^2 dx)^{1/2}, j = 1, 2$. Algorithm 3 (with $\alpha = 10^{-3}, \beta = 10^3$) performs slightly better with $\mathcal{W}_2$-distance $7.610 \times 10^{-5}$ (standard deviation: $3.367 \times 10^{-5}$) and $L^2$-distance 0.608 (standard deviation: 0.194), while Algorithm 4 has $\mathcal{W}_2$-distance $7.771 \times 10^{-5}$ (standard deviation: $3.32 \times 10^{-5}$) and $L^2$-distance 0.6434 (standard deviation: 0.451). In addition, CWB has $\mathcal{W}_2$-distance 0.0022 (standard deviation: $5.85 \times 10^{-4}$) and $L^2$-distance 0.082 (standard

Figure 3: Illustration of Wasserstein barycenters computed using WDHA, CWB and DSB.

deviation: 0.039). DSB achieves $\mathcal{W}_2$-distance $1.225 \times 10^{-5}$ (standard deviation: $2.644 \times 10^{-5}$) and $L^2$-distance 0.0699 (standard deviation: 0.034). DSB performs the best in Wasserstein distance for this example.

## C.3  $L^2$-DESCENT $\mathbb{H}^1$-ASCENT ALGORITHM

Let $\nu(x)$, $\mu_i(x)$ be density functions, we can write

$$\mathcal{J}(\nu, \phi) = \frac{1}{n} \sum_{i=1}^{n} \int_{\Omega} \left( \frac{\|x\|_2^2}{2} - \varphi_i(x) \right) \nu(x) \, \mathrm{d}x + \int_{\Omega} \left( \frac{\|x\|_2^2}{2} - \varphi_i^* \right) \mu_i(x) \, \mathrm{d}x.$$

Let $L^2(\lambda) = \{h : \int h(x)^2 \mathrm{d}\lambda < \infty\}$ be the space of functions that are $L^2$-integrable with respect to the Lebesgue measure $\lambda$ and $\mathcal{D} = \{f \in L^2(\lambda) : f(x) \geq 0, \int f(x) \mathrm{d}\lambda = 1\}$ be the set of density functions. Notice that the $L^2$-gradient of $\mathcal{J}$ with respect to $\nu(x)$ is $\boldsymbol{\nabla}_\nu \mathcal{J}(\nu, \phi) :=$

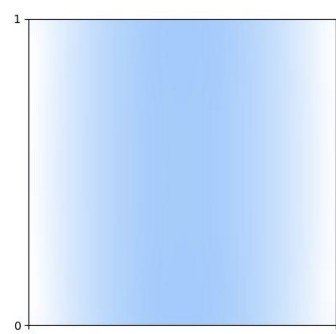

Figure 4: Plot of output from Algorithm 5 to uniform distributions on round disks.

$\frac{1}{n} \sum_{i=1}^{n} \frac{\|x\|_2^2}{2} - \varphi_i(x)$, we may consider the $L^2$-descent $\dot{\mathbb{H}}^1$-ascent algorithm described below

---

**Algorithm 5:** $L^2$-Descent $\dot{\mathbb{H}}^1$-Ascent Algorithm

---

Initialize $\nu^1, \boldsymbol{\varphi}^1$;
**for** $t = 1, 2, \cdots, T - 1$ **do**
    **for** $i = 1, 2, \ldots, n$ **do**
        $\widehat{\varphi}_i^{t+1} = \varphi_i^t + \eta_i^t \boldsymbol{\nabla}_{\varphi_i} \mathcal{J}(\nu^t, \boldsymbol{\varphi}^t)$;
        $\varphi_i^{t+1} = (\widehat{\varphi}_i^{t+1})^{**}$;
    **end**
    $\tilde{\nu}^{t+1} = \nu^t - \tau_t \boldsymbol{\nabla}_{\nu} \mathcal{J}(\nu, \phi)$;;
    $\nu^{t+1} = \mathcal{P}_{\mathcal{D}}(\tilde{\nu}^{t+1})$;
**end**
**return** $\{(\nu^t, \boldsymbol{\varphi}^t)\}_{t=1}^T$;

---

Here, $\mathcal{P}_{\mathcal{D}}(\tilde{\nu}^{t+1})$ is the projection of $\tilde{\nu}^{t+1}$ onto $\mathcal{D}$, which can be computed using Python function `pyproximal.Simplex` in the library PyProximal. We apply this algorithm to the uniform distributions supported on round disks, and observe that it diverges leading to a wrong result. Nevertheless, we plot the output in Figure 4.

# D   ADDITIONAL DETAILS

## D.1   WASSERSTEIN GRADIENT

To compute $\mathbb{W}\mathcal{J}(\nu, \varphi)$, we apply the definition in subsection 2.3. Note that

$$\frac{d}{d\varepsilon} \mathcal{J}(\nu + \varepsilon\chi, \phi)\Big|_{\varepsilon=0}$$

$$= \frac{d}{d\varepsilon} \left\{ \frac{1}{n} \sum_{i=1}^{n} \int_\Omega \frac{\|x\|_2^2}{2} - \varphi_i(x) \, d(\nu + \varepsilon\chi) + \int_\Omega \frac{\|x\|_2^2}{2} - \varphi_i^* \, d\mu_i(x) \right\}\Big|_{\varepsilon=0}$$

$$= \frac{d}{d\varepsilon} \left\{ \varepsilon \frac{1}{n} \sum_{i=1}^{n} \int_\Omega \frac{\|x\|_2^2}{2} - \varphi_i(x) \, d\chi \right\}\Big|_{\varepsilon=0}$$

$$= \frac{d}{d\varepsilon} \left\{ \varepsilon \int_\Omega \frac{\|x\|_2^2}{2} - \overline{\varphi}(x) \, d\chi \right\}\Big|_{\varepsilon=0}$$

$$= \int_\Omega \frac{\|x\|_2^2}{2} - \overline{\varphi}(x) \, d\chi,$$

which implies that $\frac{\delta\mathcal{J}}{\delta\mu}(\mu)(x) = \frac{\|x\|_2^2}{2} - \overline{\varphi}(x)$. By the definition of Wasserstein gradient, $\mathbb{W}\mathcal{J} = \nabla \frac{\delta\mathcal{J}}{\delta\mu}(\mu) = id - \nabla\overline{\varphi}$.

## D.2 IMPLEMENTATION

Let $\Omega = [0,1]^2$ and $\{(x_i, y_j)\}_{i,j=0}^m$ be the equally spaced grid points, we have $x_0 = 0, y_0 = 0$ and $x_i = i/m, y_j = j/m$ for $i, j \neq 0$. Given the evaluations $\{\varphi_{i,j} = \varphi(x_i, y_j)\}$ of function $\varphi$ on these grid points, we compute the gradient at point $(x_i, x_j), i, j \neq 0$ as

$$\nabla \varphi(x_i, y_j) = \begin{pmatrix} \frac{\varphi_{i,j} - \varphi_{i-1,j}}{h} \\ \frac{\varphi_{i,j} - \varphi_{i,j-1}}{h} \end{pmatrix},$$

where $h = 1/m$. For the computation of convex conjugates of $\varphi$, we note that for the 1D case, convex conjugate can be computed efficiently using the method in Corrias (1996). For the 2D case, notice that

$$\varphi^*(y_1, y_2) := \sup_{x_1, x_2} (x_1 - y_1)^2/2 + (x_2 - y_2)^2/2 - \varphi(x_1, x_2)$$

$$= \sup_{x_1} \left( (x_1 - y_1)^2/2 + \sup_{x_2} \left\{ (x_2 - y_2)^2/2 - \varphi(x_1, x_2) \right\} \right)$$

$$= \sup_{x_1} \left( (x_1 - y_1)^2/2 + [\varphi(x_1, \cdot)]^*(y_2) \right).$$

Thus, the convex conjugate in the 2D case can be computed by iteratively applying the 1D solver to each row and column. To discuss the implementation of $(\nabla \varphi)_{\#} \nu$, we describe how the mass $\nu(x_i, y_j)$ (density value of $\nu$ at a point $(x_i, y_j)$) is splitted and mapped (Jacobs & Léger, 2020) as follows. Since $\varphi$ is convex, we observe that $\nabla_x \varphi(x_i, y_j) \leq \nabla_x \varphi(x_{i+1}, y_j)$ and $\nabla_y \varphi(x_i, y_j) \leq \nabla_y \varphi(x_i, y_{j+1})$. Let $\mathcal{R}(x_i, y_j)$ be the quadrilateral formed by 4 points $\nabla \varphi(x_i, y_j), \nabla \varphi(x_{i+1}, y_j), \nabla \varphi(x_i, y_{j+1}), \nabla \varphi(x_{i+1}, y_{j+1})$ and pick the mesh grids $\{(\widetilde{x}_{i'}, \widetilde{y}_{j'})\}_{i',j'=1}^k \subset \mathcal{R}(x_i, y_j)$, where

$$(\widetilde{x}_{i'}, \widetilde{y}_{j'}) = (1 - \alpha_{i'})(1 - \beta_{j'}) \nabla \varphi(x_i, y_j) + \alpha_{i'}(1 - \beta_{j'}) \nabla \varphi(x_{i+1}, y_j)$$
$$+ (1 - \alpha_{i'}) \beta_{j'} \nabla \varphi(x_i, y_{j+1}) + \alpha_{i'} \beta_{j'} \nabla \varphi(x_{i+1}, y_{j+1})$$

with $0 = \alpha_0 \leq \alpha_1 \leq \cdots \leq \alpha_k = 1, 0 = \beta_0 \leq \beta_1 \leq \cdots \leq \beta_k = 1$. The mass of $\nu(x_i, y_j)$ is first uniformly distributed to the meshed grids $\{(\widetilde{x}_{i'}, \widetilde{y}_{j'})\}_{i',j'=1}^k$. Then, the mass at each point $(\widetilde{x}_{i'}, \widetilde{y}_{j'})$ is distributed to 4 nearest grid points in $\{(x_i, y_j)\}$, inversely proportional to their distances. If $(\nabla \varphi)_{\#} \nu$ exceeds the grid specified, the mass will be distributed to the boundary points instead.

