# OpenReview forum: "OPTIMAL TRANSPORT BARYCENTER VIA NONCONVEX CONCAVE MINIMAX OPTIMIZATION"
_ICLR.cc/2025/Conference — Submitted to ICLR 2025_

### Official Review · Reviewer_Pow6 · 2024-10-17

**Soundness:** 2
**Presentation:** 3
**Contribution:** 3
**Rating:** 5
**Confidence:** 4

**Summary:**

This paper uses a primal-dual formulation for the Wasserstein barycenter problem without any regularization. The method is performed by using Wasserstein gradinet descent for the optimization on the density, and for the Kantarovich potential, a H^1 metric optimization is used. The algorithm is a first order method which seems to quickly converge to the stationary point.

**Strengths:**

1. The presentation is clear, especially on the optimization for the Wasserstein gradient.

2. The numerical experiments seem promising.

**Weaknesses:**

The reviewer first wishes to comment that the methodology seems an exciting and promising direction in OT and an illuminating extension of the back-and-forth OT algorithm. However, it is hard to take the writing at face value to know the validity of the result. While the reviewer mainly asks for clarification in the remainder of this part, the unclarity of the implementation details in itself is a major weakness.

The major drawback of this work is that the methodology is not clear from the paper, which is why currently it does not seem possible to assess the quality of this work. Specifically, here are some questions that need to be answered:
1. How is the differentiation implemented in this work? Terms such as $\nabla \phi$ are not continuous and admits discrete proxies. Is it done by finite difference? And, if so, what is the scheme? Or, alternatively, is it done by performing FFT, and is some smoothing filter performed?
2. How is the pushforward implemented in this work? Assuming $\nabla \phi$ is computed, the term $(\nabla \phi)_{#}$ might exceed the grid specified. How is this going to be resolved in the numerical example? These are questions that need to be addressed within this work. As of now, the reviewer's understanding is that this work only quotes the Jacobs & Leger paper, but that would be insufficient.
3. It is quite confusing as to why the authors use a 1024*1024 grid for Sinkhorn in Figure 1. From the plot, it is clear that the distributions $\mu_1, \ldots \mu_4$ occupy a small region of the grid. Is a smaller size used? This should a priori be mentioned in the numerical experiment section. If somehow the full 1024*1024 space is used, then it would be quite unfair to the Sinkhorn algorithm. Alternatively, does the code for WDHA use the sparsity of the source and target distribution?
4. The comparison between DSB and the proposed method is still inconclusive. It may happen that numerical optimization in addition to a smaller regularization number may lead to better performance in the barycenter functional value and better wall-clock time. The reviewer is not sure if the POT code package result can be taken at face value.

Other major issues:
- The experiments primarily focus on Algorithm 4, but the analysis is done for Algorithm 3. The authors did not clearly show this intricacy in the contribution subsection.
- There does not seem to be an implementation for Algorithm 3.

Minor issues:
- This work does not sufficiently mention the limitation that it is only efficient for low-dimensional grids. Sinkhorn, in contrast, can work by discretization of continuous densities by empirical distributions. The omission to stress this limitation needs to be resolved.

**Questions:**

The questions are listed in the weakness section.

---

> ### Author Response · Authors · 2024-11-24
> **Response to Reviewer Pow6**
>
> We sincerely appreciate your time, thoughtful consideration, and constructive guidance on our paper. Following your and the other referees' valuable comments, we added three additional simulation studies in the appendix of the revised version to address your questions. If we have successfully addressed some of your concerns, we would kindly ask you to consider reflecting this in your rating. Thank you again for your invaluable feedback.
>
> Below we provide point-by-point response to your comments in italics.
>
> *1. How is the differentiation implemented in this work? Terms such as are not continuous and admit discrete proxies. Is it done by finite difference? And, if so, what is the scheme? Or, alternatively, is it done by performing FFT, and is some smoothing filter performed?*
>
> **Ans**: you are right. We used finite difference approach. In particular, let $\Omega = [0,1]^2$ and $\\{(x\_i, y\_j)\\}\_{i,j=0}^{m}$ be the $m^2$ equally spaced grid points, we have $x\_0=0, y\_0=0$ and $x\_i = i/m, y_j = j/m$ for $i,j\neq 0$. Given the evaluations $\{ \varphi\_{i,j} = \varphi(x\_i, y\_j) \}$ of function $\varphi$ on these grid points, we compute the gradient at point $(x\_i, x\_j), i, j \neq 0$ as
> \begin{align*}
> \nabla \varphi (x_i, y_j) =
> \left(
> \begin{array}{c}
>     \frac{\varphi_{i,j} - \varphi_{i-1, j}}{h}   \\
>       \frac{\varphi_{i,j} - \varphi_{i, j-1}}{h}
> \end{array}
> \right),
> \end{align*}
> where $h=1/m$. No smoothing or filtering is performed for computing the gradient. FFT is used to solve the Poisson equation ($\Delta f = u$) with zero Neumann boundary condition.
>
> *2. How is the pushforward implemented in this work? Assuming is computed, the term $(\nabla \phi)\_{\\#}$ might exceed the grid specified. How is this going to be resolved in the numerical example? These are questions that need to be addressed within this work. As of now, the reviewer's understanding is that this work only quotes the Jacobs \& Leger paper, but that would be insufficient.*
>
> **Ans**: To discuss the implementation of $(\nabla \varphi)\_{\\#} \nu$, we describe how the mass $\nu(x\_i, y\_j)$ (density value of $\nu$ at a point $ (x\_i, y\_j)$) is splitted and mapped \citep{jacobs2020fast} as follows. Since $\varphi$ is convex, we observe that $\nabla\_{x} \varphi (x\_i, y\_j) \leq \nabla\_{x} \varphi (x\_{i+1}, y\_j)$ and $\nabla\_{y} \varphi (x\_i, y\_j) \leq \nabla\_{y} \varphi (x\_{i}, y\_{j+1})$. Let $\mathcal{R}(x\_i,y\_j)$ be the quadrilateral formed by 4 points $\nabla\varphi(x\_i,y\_j),\nabla\varphi(x\_{i+1},y\_j),\nabla\varphi(x\_i,y\_{j+1}),\nabla\varphi(x\_{i+1},y\_{j+1})$ and pick the mesh grids $\\{ (\widetilde{x}\_{i'}, \widetilde{y}\_{j'}) \\}\_{i',j'=1}^k \subset \mathcal{R}(x\_i,y\_j)$, where
> $$
> (\widetilde{x}\_{i'}, \widetilde{y}\_{j'}) =  (1 - \alpha\_{i'})(1 - \beta\_{j'}) \nabla\varphi(x\_i, y\_j) + \alpha\_{i'}(1 - \beta\_{j'}) \nabla\varphi(x\_{i+1}, y\_j) + (1 - \alpha\_{i'})\beta\_{j'} \nabla\varphi(x\_i, y\_{j+1}) + \alpha\_{i'} \beta\_{j'} \nabla\varphi(x\_{i+1}, y\_{j+1})
> $$
> with $0 = \alpha_{0}\leq \alpha_1 \leq \dots \leq \alpha_{k} = 1, 0 = \beta_{0}\leq \beta_1 \leq \dots \leq \beta_{k} = 1 $. The mass of $\nu(x\_i,y\_j)$ is first uniformly distributed to the meshed grids $\\{(\widetilde{x}\_{i'}, \widetilde{y}\_{j'})\\}\_{i',j'=1}^k$. Then, the mass at each point $ (\widetilde{x}\_{i'}, \widetilde{y}\_{j'}) $ is distributed to 4 nearest grid points in $\\{ (x\_i, y\_j )\\}$, inversely proportional to their distances. If $(\nabla \varphi)\_{\\#} \nu$ exceeds the grid specified, the mass will be distributed to the boundary points instead. The discussions about computing $\nabla \varphi$, $\varphi^{*}$ and $(\nabla \phi)_{\\#} \nu $ are added to appendix subsection D.2 of the revised version.
>
>
> *3. It is quite confusing as to why the authors use a 1024 $\times$ 1024 grid for Sinkhorn in Figure 1. From the plot, it is clear that the distributions occupy a small region of the grid. Is a smaller size used? This should a priori be mentioned in the numerical experiment section. If somehow the full 1024 $\times$ 1024 space is used, then it would be quite unfair to the Sinkhorn algorithm. Alternatively, does the code for WDHA use the sparsity of the source and target distribution?*
>
> **Ans**: our method is advantageous for high-resolution densities. The $1024 \times 1024$ is the standard pixel size for images and has nothing in particular. Our method wins on $500 \times 500$ images (demonstrated by example 2) as well. All the methods are compared on the full grid and are fed with the same densities, and thus the comparison is fair. In example 1, we put four shapes in the corners solely for visualization purpose, we can enlarge and center the shapes, the results would be the same. Example 2 would correspond to the case that the distributions occupy large regions around the center.

---

> > ### Author Response · Authors · 2024-11-24
> > **Response to Reviewer Pow6**
> >
> > *4. The comparison between DSB and the proposed method is still inconclusive. It may happen that numerical optimization in addition to a smaller regularization number may lead to better performance in the barycenter functional value and better wall-clock time. The reviewer is not sure if the POT code package result can be taken at face value.*
> >
> > **Ans**: We added a new simulation in the revised version for which the ground truth is known. We emphasize that regularization parameters reg= 0.003 and reg=0.005 are the **optimal** choices for CWB and DSB respectively. Smaller regularization parameters lead nonconvergent and worse results for both CWB and DSB, please see the table below and Figure 3 of the revised version. With known ground truth, we can compare directly the Wasserstein distance between computed barycenter and the truth, and $L^2$-distance between the computed barycenter density and the true density. This simulation demonstrates that both CWB and DSB may output blurry barycenters, while our result is visually no different from the truth.
> >
> > In this new simulation, the goal is to compute the barycenter of four uniform distributions supported on round disks of radius 0.15, centered at $(0.2, 0.2)$, $(0.2, 0.8)$, $(0.8, 0.2)$, $(0.8, 0.8)$ respectively. It's clear that the true barycenter is uniform on the disk of radius 0.15 centered at $(0.5, 0.5)$. The computed barycenter densities by WDHA, CWB and DSB are shown in Figure 3 of the revised version. We report in the following table the Wasserstein distance between computed barycenter distribution to the truth, $L^2$-distance between computed barycenter densities and the true density, and the barycenter functional value. Our method is uniformly the best, and in particular, the improvement in the Wasserstein distance is of orders of magnitude.
> >
> > |               |   reg     | Wasserstein distance      |    $L^2$-distance     | $\mathcal{F}(\nu^{est})$   |
> > |:----------|:--------:|:----------------------------:|:-----------------------:|:-----------------------------:|
> > |WDHA   |             |    $3.758\times 10^{-8}$   |  0.4869                     | $9.001 \times 10^{-2}$    |
> > |CWB     |   0.003 |    $2.98\times 10^{-4}$    | 1.321                        | $9.031 \times 10^{-2}$      |
> > | CWB    |  0.002  |    $2.538 \times 10^{-2}$  | 3.041                       |            0.1154                     |
> > |DSB      | 0.005   | $1.2 \times 10^{-4}$         | 0.8219                     | $9.013\times 10^{-2}$       |
> > |   DSB   | 0.004.  | $1.164\times 10^{-2}$.     | 2.281                       | 0.1016                                |
> >
> > We mark that the barycenter functional value of the truth is 0.09. This simulation is included in the appendix subsection C.1 of the revised version. Our code is attached for reproducibility check.

---

> > > ### Author Response · Authors · 2024-11-24
> > > **Response to Reviewer Pow6**
> > >
> > > *5. The experiments primarily focus on Algorithm 4, but the analysis is done for Algorithm 3. The authors did not clearly show this intricacy in the contribution subsection. There does not seem to be an implementation for Algorithm 3.*
> > >
> > > **Ans**: Thank you. Empirically, the projection can be computed by the approach in [Simonetto, 2021], which is implemented as the python function `reg4opt.regression.convex\_regression` in library `reg4opt`. Unfortunately, this requires solving a large  convex quadratically constrained quadratic problem, and the code could take weeks to run.
> > >
> > > Instead, we implement the algorithm with projection (Algorithm 3) and test the method on 1D distributions. For each repetition $t$ and $i=1,2,3$, we let $\mu\_{i,t}$ be the truncated version of $N(a\_i,{\sigma\_{i}}^2)$ on the domain $[0,1]$, where $a\_i \sim \text{uniform}\left[0.3,0.7\right]$ and  $\sigma\_i \sim \text{uniform}\left[0,1\right]$. Let $\overline{\nu}_t$, $\overline{\nu}\_{1, t}$, $\overline{\nu}\_{2, t}$
> > > be the true barycenter, computed barycernter from Algorithm 3 (with projection onto $\mathbb{F}\_{\alpha, \beta}$), computed barycernter from Algorithm 4 (with double convex conjugates) respectively. We repeat the experiment 300 times and report two types of average distances between the groundtruth and each computed barycenters : average $\mathcal{W}\_2$-distance $\frac{1}{T} \sum\_{t=1}^T \mathcal{W}\_2(\overline{\nu}\_t, \overline{\nu}\_{j, t})$ and average $L^2$-distance between densities $ \frac{1}{T} \sum\_{t=1}^T (\int (\overline{\nu}\_t(x)- \overline{\nu}\_{j, t}(x))^2 d x)^{1/2}, j=1,2$. Algorithm 3 performs slightly better with $\mathcal{W}\_2$-distance $7.610\times 10^{-5}$ and $L^2$-distance 0.608, while Algorithm 4 has $\mathcal{W}_2$-distance $7.771\times 10^{-5}$ and $L^2$-distance 0.6434. We have included this study in subsection C.2 in the appendix of the revised version.
> > >
> > > We remark that the same issue also exists in other works that utilize the $\mathbb{H}^1$-gradient if any convergence theories are to be developed; see the assumptions  in proposition 1 [Jacobs and L´eger, 2020] and Theorem 2.8 [Jacobs et al., 2021]. We don't think this should be the reason to deny its satisfying empirical performance either. Please check our additional simulation with ground truth. Without additional regularities, generalization of the Theorem can be challenging, the main obstacle is that the map $\nu \rightarrow T_{\nu}^{\mu}$ is not Lipschitz, see [Berman, 2021] for more details regarding this.

---

> > > > ### Comment · Reviewer_Pow6 · 2024-11-24
> > > >
> > > > The reviewer thanks the author for the helpful clarification. The reviewer has decided to maintain the score on the basis of two significant weaknesses: (A) The structure of the manuscript focuses too much on the less practical Algorithm 3, and the Algorithm 4 is less developed as it should be, and (B) The numerical is not as strong as that of Jacob & Leger. For (A), the detailed comment is that this factor is a major issue in the presentation, as the readers would read into Section 4 thinking that the Algorithm has a convergence guarantee, but in fact the presented algorithm lacks so. For (B), the main issue is that the WDHA Wasserstein barycenter does look worse than the WSB, and it looks worse in the sense that the barycenter looks quite nonsmooth, which begs the question of if the wasserstein gradient outer loop proposed in this work is sound, or if better algorithms can be used. In practice, one would pass the WSB result over a sharpening filter or pass the DSB result over a smoothing filter. The issue is that the WDHA result (i) doesn't look good, (ii) the $\nu$ will have a Soblev norm that is excessively large, and (iii) doesn't have a ground truth that can convince the readers of the validity of a plot that doesn't look good and has bad Sobolev norm.
> > > >
> > > > The reviewer would have given the work a score of 4 were it an option, while the reviewer thinks the omission of technical details were probably minor omissions and giving a score of 3 would be quite off in the reviewer's assessment, which was the reasoning for choosing 5. After the clarification, the reviewer thinks the work warrants a 5, but higher than that grade would not be possible due to issues (A) and (B). Fixing either (A) or (B) in the reviewer's opinion could very well lead to a 6, and fixing both will make the work very good. The reviewer thinks the work is nevertheless quite promising and wishes the authors the best of luck.

---

> ### Author Response · Authors · 2024-11-25
> **Response to Reviewer Pow6**
>
> We just want to add the following points.
>
> *(iii) doesn't have a ground truth that can convince the readers of the validity of a plot that doesn't look good and has bad Sobolev norm*
>
> **Ans**: we added a new simulation in the appendix  subsection C.1 which has gournd truth. The ground truth of this example is the uniform distribution on the disk of radius 0.15 centered at (0.5, 0.5), which is not smooth and has $\dot{\mathbb{H}}^{-1}$-norm (negative Sobolev norm) bounded by 1. Please kindly see our response to point 4.
>
> *(i) doesn't look good*
>
> **Ans**: In our newly added simulation example in the appendix of subsection C.1, the Wasserstein distance between the barycenter of our method and the truth is smaller than that of both CWB and DSB. The $L^2$-distance between the density of barycenter of our method and the truth is also the smallest. We think that our barycenter look better visually.
>
> *In practice, one would pass the WSB result over a sharpening filter or pass the DSB result over a smoothing filter.*
>
> **Ans**: we indeed conducted an additional thresholding step for WSB and CSB in exmaple 1. Interested readers can check Figure 1 of the paper for the results.

---

### Official Review · Reviewer_BsSs · 2024-11-01

**Soundness:** 3
**Presentation:** 3
**Contribution:** 3
**Rating:** 6
**Confidence:** 3

**Summary:**

This paper introduces a new algorithm called the "Wasserstein-Descent H-Ascent (WDHA) algorithm" for computing the optimal transport barycenter.
Considering the dual form of the original problem, this paper transforms the barycenter problem as   a nonconvex-concave minimax optimization problem, and then propose a gradient algorithm for optimization.
One advantage of the proposed WDHA algorithm is its runtime complexity, which grows linearly with the grid size $m$, while the naive method computing the optimal transport map requires time complexity of $O(m^3)$.
Moreover, the under certain assumptions, the paper also shows the convergence rate $O(1/T)$ of the algorithm to  its stationary point.
Finally, numerical studies are provided to demonstrate the effectiveness of the method.

**Strengths:**

- Algorithm: The paper introduces the WDHA algorithm by considering the dual form of the problem of the optimal transport barycenter and using a  nonconvex-concave minimax optimization procedure, which is novel and interesting.
Moreover, The proposed WDHA algorithm achieves nearly linear runtime complexity, improving the $O(m^3)$ complexity of the naive approach.

- Theory: Under certain assumptions, the authors provide a convergence rate of  for the WDHA algorithm, which matches the convergence rate as in the Euclidean nonconvex-concave optimization problems.

- Numerical study: The paper includes numerical studies that demonstrate the effectiveness of the WDHA algorithm over existing methods, showing that the proposed algorithm runs faster than the existing methods.

**Weaknesses:**

- Theory: One weakness is that the theoretical analysis is performed for Algorithm 3, while the practical implementation uses Algorithm 4.
Also, the theorem assumes lower bounded densities of the distributions, which may not be true in practice (for example, in the case of Figure 1). Even in this ideal setting, there is also an extra assumption on the uniform boundedness of $\nu^t$. Therefore, the convergence guarantee of the algorithm in practice remains unclear.

- Numerical Studies: While the numerical studies show faster computation times for the proposed method, I think the current results are not sufficiently convincing to demonstrate its superiority. For example, the "barycenter functional values" reported in line 515 differ very slightly across differet methods, and it seems to me that in the figures the DSB method performs generally well. More quantitative and comprehensive experiments should be conducted to validate the claims.

  Moreover, I would like to ask the authors to provide the codes for reproducibility.

- Comparison with Literature: A detailed discussion and comparison regarding the settings, runtime complexity, and theoretical guarantees with existing methods for optimal transport barycenter computation should be included for better demonstration.

**Questions:**

1. Can you provide theoretical/empirical insights for the statement in Line 317 "By embedding all densities functions into L2(Ω), we can alternatively use the L2-gradient to update ν. However, in simulations, the Wasserstein gradient update significantly outperforms the L2-gradient update."

2. Can you discuss the impact of the additional regularity by considering $\mathbb{F}_{\alpha,\beta}$ on the resulting barycenter accuracy, the convergence rate and computational cost?
How these quantities are chosen in practice?

3. I would like the authors to discuss more on the effect of tuning the parameters in the numerical studies.

---

> ### Author Response · Authors · 2024-11-24
> **Response to Reviewer BsSs**
>
> We sincerely appreciate your time, thoughtful consideration, and constructive guidance on our paper. Following your and the other referees' valuable comments, we added three additional simulation studies in the appendix of the revised version to address your questions. If we have successfully addressed some of your concerns, we would kindly ask you to consider reflecting this in your rating. Thank you again for your invaluable feedback.
>
> Below we provide point-by-point response to your comments in italics.
>
> *1. Theory: One weakness is that the theoretical analysis is performed for Algorithm 3, while the practical implementation uses Algorithm 4. Also, the theorem assumes lower bounded densities of the distributions, which may not be true in practice (for example, in the case of Figure 1). Even in this ideal setting, there is also an extra assumption on the uniform boundedness of $\nu^t$. Therefore, the convergence guarantee of the algorithm in practice remains unclear.*
>
> **Ans**: Thank you. The same issue also exists in other works that utilize the $\mathbb{H}^1$-gradient if any convergence theories are to be developed; see the assumptions  in proposition 1 [Jacobs and L´eger, 2020] and Theorem 2.8 [Jacobs et al., 2021]. We don't think this should be the reason to deny its satisfying empirical performance either. Please check our additional simulation with ground truth. Without additional regularities, generalization of the Theorem can be challenging, the main obstacle is that the map $\nu \rightarrow T\_{\nu}^{\mu}$ is not Lipschitz, see [Berman, 2021] for more details regarding this.
>
> Empirically, the projection can be computed by the approach in [Simonetto, 2021], which is implemented as the Python function `reg4opt.regression.convex\_regression` in library `reg4opt`. Unfortunately, this requires solving a large  convex quadratically constrained quadratic problem, and the code could take weeks to run. Instead, we implement the algorithm with projection (Algorithm 3) and test the method on 1D distributions. For each repetition $t$ and $i=1,2,3$, we let $\mu\_{i,t}$ be the truncated version of $N(a\_i,{\sigma\_{i}}^2)$ on the domain $[0,1]$, where $a\_i \sim \text{uniform}\left[0.3,0.7\right]$ and  $\sigma\_i \sim \text{uniform}\left[0,1\right]$. Let $\overline{\nu}\_t$, $\overline{\nu}\_{1, t}$, $\overline{\nu}\_{2, t}$
> be the true barycenter, computed barycernter from Algorithm 3 (with projection onto $\mathbb{F}\_{\alpha, \beta}$), computed barycernter from Algorithm 4 (with double convex conjugates) respectively. We repeat the experiment 300 times and report two types of average distances between the groundtruth and each computed barycenters : average $\mathcal{W}\_2$-distance $\frac{1}{T} \sum\_{t=1}^T \mathcal{W}\_2(\overline{\nu}\_t, \overline{\nu}\_{j, t})$ and average $L^2$-distance between densities $ \frac{1}{T} \sum\_{t=1}^T (\int (\overline{\nu}\_t(x)- \overline{\nu}\_{j, t}(x))^2 d x)^{1/2}, j=1,2$. Algorithm 3 performs slightly better with $\mathcal{W}\_2$-distance $7.610\times 10^{-5}$ and $L^2$-distance 0.608, while Algorithm 4 has $\mathcal{W}\_2$-distance $7.771\times 10^{-5}$ and $L^2$-distance 0.6434. We have included this study in subsection C.2 in the appendix of the revised version.
>
> In example 1, we put four shapes in the corners solely for visualization purpose, we can enlarge and center the shapes, the results would be the same. In addition, it's possible to separate the domains of the measures by following precedures in Chartrand et al. [2009] and considering the following formulation
> $$
> \mathcal{I}\_{\nu}^{\mu\_i}(\varphi\_i) = \int\_\Omega\frac{\|x\|\_2^2}{2} - \varphi\_i(x) d \nu + \int\_{\Phi\_i} \frac{\|x\|\_2^2}{2} - \varphi\_i^\ast d \mu\_i(x),
> $$
> where the support of $\nu$ is in $\Omega$, the suport of $\mu\_i$ is in $\Phi\_i$ and $\varphi\_i : \Omega \rightarrow \mathbb{R}$. The convex conjugate of $\varphi$ is defined to be $\varphi^{*}(y) = \sup\_{x \in \Omega} x^T y - \varphi(x)$ for any $y \in \Phi\_i$. It's then possible to extend the results by using Proposition 2.1 and Lemma 2.2 in  Chartrand et al. [2009]. Then, $\mu\_i$ will be bounded only on $\Phi\_i$.
>
> We note that the boundedness assumption of $\nu^t$ is minor, because this can be easily checked in practice. If this assumption is violated, a warning message can be sent to the user.

---

> > ### Author Response · Authors · 2024-11-24
> > **Response to Reviewer BsSs**
> >
> > *2. Numerical Studies: While the numerical studies show faster computation times for the proposed method, I think the current results are not sufficiently convincing to demonstrate its superiority. For example, the ``barycenter functional values" reported in line 515 differ very slightly across different methods, and it seems to me that in the figures the DSB method performs generally well. More quantitative and comprehensive experiments should be conducted to validate the claims. Moreover, I would like to ask the authors to provide the codes for reproducibility.*
> >
> > **Ans**:  that's a good point. We added a new simulation in the revised version for which the ground truth is known. So, we can compare directly the Wasserstein distance between the computed barycenter and the truth, and $L^2$-distance between the computed barycenter density and the true density. This simulation demonstrates that both CWB and DSB may output blurry barycenters, while our result is visually no different from the truth.
> >
> > In this new simulation, the goal is to compute the barycenter of four uniform distributions supported on round disks of radius 0.15, centered at $(0.2, 0.2)$, $(0.2, 0.8)$, $(0.8, 0.2)$, $(0.8, 0.8)$ respectively. It's clear that the true barycenter is uniform on the disk of radius 0.15 centered at $(0.5, 0.5)$. The computed barycenter densities by WDHA, CWB and DSB are shown in Figure 3 of the revised version. We note that regularization parameters reg= 0.003 and reg=0.005 are the optimal choices for CWB and DSB respectively. Smaller regularization parameters lead nonconvergent and worse results for both CWB and DSB. We report in the following table the Wasserstein distance between computed barycenter distribution to the truth, $L^2$-distance between computed barycenter densities and the true density, and the barycenter functional value. Our method is uniformly the best, and in particular, the improvement in the Wasserstein distance is of orders of magnitude.
> >
> > |               |   reg     | Wasserstein distance      |    $L^2$-distance     | $\mathcal{F}(\nu^{est})$   |
> > |:----------|:--------:|:----------------------------:|:-----------------------:|:-----------------------------:|
> > |WDHA   |             |    $3.758\times 10^{-8}$   |  0.4869                     | $9.001 \times 10^{-2}$    |
> > |CWB     |   0.003 |    $2.98\times 10^{-4}$    | 1.321                        | $9.031 \times 10^{-2}$      |
> > | CWB    |  0.002  |    $2.538 \times 10^{-2}$  | 3.041                       |            0.1154                     |
> > |DSB      | 0.005   | $1.2 \times 10^{-4}$         | 0.8219                     | $9.013\times 10^{-2}$       |
> > |   DSB   | 0.004.  | $1.164\times 10^{-2}$.     | 2.281                       | 0.1016                                |
> >
> > We mark that the barycenter functional value of the truth is 0.09. This simulation is included in the appendix subsection c1 of the revised version. Our code is attached for reproducibility check.
> >
> > *3.  Comparison with Literature: A detailed discussion and comparison regarding the settings, runtime complexity, and theoretical guarantees with existing methods for optimal transport barycenter computation should be included for better demonstration.*
> >
> > **Ans**: our method is advantageous for high-resolution densities, i.e., densities supported on extensive grid points. To our best knowledge, CWB and DSB are the only two methods are applicable to this setting, i.e., can have an output within reasonable time, and have available implementations. The runtime for example 1 and example 1 are reported in our paper. In addition, we have added another example with gourd truth. Please see our response for point 2. We are the first to use $\mathbb{\dot{H}}^1$-gradient in the barycenter problem. The only theoretical guarantees of optimization algorithms that use $\mathbb{\dot{H}}^1$-gradient are the ones developed in [Jacobs and L´eger, 2020, Jacobs et al., 2021], which are also based on similar regularities (smoothness and strong convexity) of potentials.

---

> > > ### Author Response · Authors · 2024-11-24
> > > **Response to Reviewer BsSs**
> > >
> > > *4. Can you provide theoretical/empirical insights for the statement in Line 317 ``By embedding all densities functions into $L^2(\Omega)$, we can alternatively use the L2-gradient to update $\nu$. However, in simulations, the Wasserstein gradient update significantly outperforms the L2-gradient update."*
> > >
> > > **Ans**: Certainly, we implemented this algorithm and tested it on uniform distributions supported on round disks for which the ground truth is known. Let $\nu(x)$, $\mu_i(x)$ be density functions. We can write
> > > $$
> > > \mathcal{J}(\nu, \phi) = \frac{1}{n} \sum\_{i=1}^n \int\_\Omega \left(\frac{\|x\|\_2^2}{2} - \varphi\_i(x) \right) \nu(x)  d x + \int\_\Omega \left( \frac{\|x\|\_2^2}{2} - \varphi\_i^\ast \right) \mu\_i(x) d x.
> > > $$
> > > Let $L^2(\lambda)=\\{ h : \int h(x)^2 d \lambda < \infty \\}$ be the space of functions that are $L^2$-integrable with respect to the Lebesgue measure $\lambda$ and $\mathcal{D} = \{ f \in L^2(\lambda): f(x) \geq 0, \int f(x) d \lambda =1 \}$ be the set of density functions. Notice that the $L^2$-gradient of $\mathcal{J}$ with respect to $\nu(x)$ is $\nabla\_{\nu} \mathcal{J}(\nu, \phi):= \frac{1}{n} \sum\_{i=1}^n \frac{\|x\|\_2^2}{2} - \varphi\_i(x)$, we may consider the $L^2$-descent $\mathbb{\dot{H}}^1$-ascent algorithm: for each iteration $t$, do
> > >
> > > - For $i=1,2, \dots, n$, compute
> > > $$
> > >     \begin{aligned}
> > >     & \widehat{\varphi}\_{i}^{t+1} = \varphi\_{i}^{t} + \eta^t\_i \nabla\_{\varphi_i} \mathcal{J} (\nu^t, \varphi^{t});
> > >     & \varphi\_{i}^{t+1} = (\widehat{\varphi}\_{i}^{t+1})^{**}
> > > \end{aligned}
> > > $$
> > > - Compute
> > > $$
> > > \begin{aligned}
> > > \tilde{\nu}^{t+1} & =  \nu^{t} - \tau\_t \nabla \_{\nu} \mathcal{J}(\nu, \phi);
> > > \nu^{t+1}  = \mathcal{P}_{\mathcal{D}}(\tilde{\nu}^{t+1})
> > > \end{aligned}
> > > $$
> > >
> > > Here, $ \mathcal{P}_{\mathcal{D}}(\tilde{\nu}^{t+1})$ is the projection of $\tilde{\nu}^{t+1}$ onto $\mathcal{D}$, which can be computed using python function `pyproximal.Simplex` in library `1PyProximal`. We apply this algorithm to the uniform distributions supported on round disks, and observe that it actually diverges leading to a wrong result. Nevertheless, we plot the output in Figure 4 of the revised version. This simulation is included in appendix subsection C.3 of the revised version.
> > >
> > > *5. Can you discuss the impact of the additional regularity by considering on the resulting barycenter accuracy, the convergence rate and computational cost? How these quantities are chosen in practice?*
> > >
> > > **Ans**: for the class of problems that the potentials between true barycenter and each $\mu\_i$ are strongly convex and smooth, we expect the additional regularities to be redundant. Without this additional regularity, the inner maximization problem is not strongly concave and one may only show convergence up to $\epsilon$-precision. This can be reflected by its Euclidean counterpart for nonconvex-concave problem, where we only have concavity, but not strong concavity, see section 4.2 of Lin et al. [2020]. Projecting the potential function to $\mathbb{F}_{\alpha, \beta}$ is expensive having $O(m^2)$ time complexities [Simonetto, 2021], and we thus recommend replacing the projection step by double convex conjugates having time complexity $O(m \log m)$.
> > >
> > > *6. I would like the authors to discuss more on the effect of tuning the parameters in the numerical studies.*
> > >
> > > **Ans**: we revised our discussion of parameters $\tau, \eta$ on page 420 in the old version to ``Theorem 1 suggests that the parameters $\tau$, $\eta$ should be bounded above. Empirically, we find that the above algorithm works better with diminishing step sizes, potentially due to two reasons: (1) diminishing step sizes may reduce the effect of discrete approximations; and (2) diminishing step sizes may be more effective for nonsmooth convex functions. Since the second convex conjugate does not enforce strong convexity and smoothness, Lemma 1 no longer holds, and $\mathcal{I}\_{\nu}^{\mu}$ is only guaranteed to be concave. In addition, diminishing step size may speed up the learning process in the early stages and inverse time decay for $\tau\_t$, i.e., $\tau\_t = 1/t$, works equally well in the simulation studies." in the revised version.

---

> > > > ### Comment · Reviewer_BsSs · 2024-11-26
> > > >
> > > > I would like to thank the authors for their detailed response and new experiments. I will raise the scores accordingly.

---

> > > > > ### Author Response · Authors · 2024-11-26
> > > > > **Response to Reviewer BsSs**
> > > > >
> > > > > We sincerely appreciate your constructive comments and time!

---

### Official Review · Reviewer_yaTH · 2024-11-03

**Soundness:** 3
**Presentation:** 3
**Contribution:** 2
**Rating:** 6
**Confidence:** 4

**Summary:**

In this paper, the authors propose a novel method to compute the unregularized Wasserstein barycenter of probability distributions discretized on compact sets. This setting is typically encountered in image settings, where one image corresponds to a probability distribution over a grid of pixels. In essence, their method consists in reformulating the Wasserstein barycenter problem as a min-max optimization problem (by relying on the dual formulation of the Kantorovitch transport problem), where the 'max'-based problem is concave while the 'min'-based problem is not convex. Then, they propose to solve this new problem via gradient ascent-descent approach, an iterative algorithm which alternates between a gradient descent step operating on the measure space (with Wasserstein gradient) and gradient ascent steps operating on the space of Kantorovitch dual potentials. Under reasonable assumptions on the input distributions, the authors provide convergence rate to a stationary point and iteration complexity of their algorithm named WDHA. Then, they apply their method to image settings (synthetic data + binarized MNIST) and compare WDHA to previous methods, which rely on regularization.

**Strengths:**

- The paper is well written, the authors paying attention to define their mathematical notation and justifying carefully their computations (especially on gradients). A valuable effort is made on introducing elements in the introduction that will serve later for defining the theoretical framework of the method. In particular, the algorithm is clear to understand.
- The numerical experiments provide the intuition that the current method provide good quality of the barycenter.

**Weaknesses:**

- The choice of numerical setting of the main hyperparameters (step-sizes for ascent and descent) is not discussed.
- The burden of dimension is not discussed: in the current paper, only 2-dimensional settings are considered while Ot problems are generally of interest in larger scale. It is hard to see if the current method is scalable.
- The authors compare their methods to regularized approaches that enable to compute Wasserstein barycenters, but did not consider certain regularized methods that provide the exact same type of numerics (barycenter over images), see [1,2], without proper justification.
- Numerical results do not include uncertainty quantifications, which hurts the statistical significance of the results.
- The current paper lacks experiments where the actual barycenter is known (for example Gaussian experiment), which would enable to have a clear comparison with ground truth samples.

[1] Fast Computation of Wasserstein Barycenters. Cuturi and Doucet. 2014.

[2] Continuous Regularized Wasserstein Barycenters. Li et al. 2020.

**Questions:**

- How much critical is the compactness assumption on the input probability distributions for your method ?
- Assumptions in Theorem 1 suggest that the input distributions should have the same support (although it is not verified in the very first experiment). Could you improve this assumption to consider a case of distinct supports ?
- Could you explain what prevents the term $\mathcal{F}_{\alpha, \beta}(\nu^{T+1})$ to be arbitrarily large in the upper bound of Theorem 1, Which may hurt the convergence rate ?
- Could you detail how you efficiently implement the computation of the convex conjugate and the negative inverse Laplacian in the case of probability density functions discretized over a grid ?
- How do your method compare with methods mentioned in the section 'Weaknesses' ?
- What motivated to set the hyperparameters of your algorithm (in particular $\tau$) as you did ? Do you have guidelines for general use ?
- How do you compute the value function $\mathcal{F}(\nu^{est})$ for competing methods ?

**Suggestions**
- The authors should mention in the introduction/related work the other common setting for Wasserstein barycenters related to generative modeling and deep learning, where the input probability distributions are not tractable and are represented by subset of samples. In the image setting, this would correspond to take one image as a sample from a high-dimensional distribution (not a distribution itself). A lot of recent work has been done in this direction recently.
- I think there is a typo at the end of page 5: it should be a max not a min.

---

> ### Author Response · Authors · 2024-11-24
> **Response to Reviewer yaTH**
>
> We sincerely appreciate your time, thoughtful consideration, and constructive guidance on our paper. Following your and the other referees' valuable comments, we added three additional simulation studies in the appendix of the revised version to address your questions. If we have successfully addressed some of your concerns, we would kindly ask you to consider reflecting this in your rating. Thank you again for your invaluable feedback.
>
> Below we provide point-by-point response to your comments in italics.
>
> *1. The choice of numerical setting of the main hyperparameters (step-sizes for ascent and descent) is not discussed. What motivated to set the hyperparameters of your algorithm (in particular $\tau\_t$) as you did? Do you have guidelines for general use?*
>
> **Ans**: We revised our discussion of parameters $\tau, \eta$ on line 420 in the old version to ``Theorem 1 suggests that the parameters $\tau$, $\eta$ should be bounded above. Empirically, we find that the above algorithm works better with diminishing step sizes, potentially due to two reasons: (1) diminishing step sizes may reduce the effect of discrete approximations; and (2) diminishing step sizes may be more effective for nonsmooth convex functions. Since the second convex conjugate does not enforce strong convexity and smoothness, Lemma 1 no longer holds, and $\mathcal{I}\_{\nu}^{\mu}$ is only guaranteed to be concave. In addition, diminishing step size may speed up the learning process in the early stages and a inverse time decay for $\tau\_t$, i.e., $\tau\_t = 1/t$, works equally well in the simulation studies." in the revised version.
>
> *2. The burden of dimension is not discussed: in the current paper, only 2-dimensional settings are considered while Ot problems are generally of interest in larger scale. It is hard to see if the current method is scalable. The authors should mention in the introduction/related work the other common setting for Wasserstein barycenters related to generative modeling and deep learning, where the input probability distributions are not tractable and are represented by subset of samples. In the image setting, this would correspond to take one image as a sample from a high-dimensional distribution (not a distribution itself). A lot of recent work has been done in this direction recently.*
>
> **Ans**: Thank you for the comment. Using deep neural networks and generative models may be the only way to break the curse of dimensionality, we have added the following discussion in the revised paper on line 58 ``To break the curse of dimensionality, generative models have been investigated for the Wasserstein barycenter problem [Korotin et al., 2022]”". Feel free to let us know if we missed any relevant references.
>
> *3. The authors compare their methods to regularized approaches that enable to compute Wasserstein barycenters, but did not consider certain regularized methods that provide the exact same type of numerics (barycenter over images), see [1,2], without proper justification. How do your method compare with methods mentioned in the section ’Weaknesses’?*
>
> **Ans**: We note that [1] is an earlier work of Marco Cuturi who also co-authored CWB and DSB, which are both compared in the paper. DSB is an fixed support method that is more efficient than free support method; see [1] for this conclusion. We tried to add [2] as a comparison method. Unfortunatly, the code in (https://github.com/lingxiaoli94/CWB/tree/master) is not directly applicable to our examples, we thus provide the following justification on line 55 in the revised version ``Also, Li et al. [2020]  propose a new dual formulation for the regularized Wasserstein barycenter problem such that discretizing the support is not needed.". Since both methods rely on entropic regularizations, the issue of having blurry output is still expected to exist.
>
> [1] Fast Computation of Wasserstein Barycenters. Cuturi and Doucet. 2014.
> [2] Continuous Regularized Wasserstein Barycenters. Li et al. 2020.
>
> *4. Numerical results do not include uncertainty quantifications, which hurts the statistical significance of the results.*
>
> **Ans**: We added a new exmaple for 1D distributions to compare the algorithm with projection (Algorithm 3) and the algorithm with double convex conjugates (Algorithm 4). In this simulation, we repeat the experiments 300 times and report the average distance to address uncertainty quantification. Please see subsection C.2 in the appendix of the revised version for more details.

---

> > ### Author Response · Authors · 2024-11-24
> > **Response to Reviewer yaTH**
> >
> > *5. The current paper lacks experiments where the actual barycenter is known (for example Gaussian experiment), which would enable to have a clear comparison with ground truth samples.*
> >
> > **Ans**: That's a good point. We added a new simulation in the revised version for which the ground truth is known. This simulation demonstrates that both CWB and DSB may output blurry barycenters, while our result is visually no different from the truth.
> >
> > In this new simulation, the goal is to compute the barycenter of four uniform distributions supported on round disks of radius 0.15, centered at $(0.2, 0.2)$, $(0.2, 0.8)$, $(0.8, 0.2)$, $(0.8, 0.8)$ respectively. It's clear that the true barycenter is uniform on the disk of radius 0.15 centered at $(0.5, 0.5)$. The computed barycenter densities by WDHA, CWB and DSB are shown in Figure 3 of the revised version. We note that regularization parameters reg= 0.003 and reg=0.005 are the optimal choices for CWB and DSB respectively. Smaller regularization parameters lead to nonconvergent and worse results for both CWB and DSB. We report in the following table the Wasserstein distance between computed barycenter distribution to the truth, $L^2$-distance between computed barycenter densities and the true density, and the barycenter functional value. Our method is uniformly the best, and in particular, the improvement in the Wasserstein distance is of orders of magnitude.
> >
> > |               |   reg     | Wasserstein distance      |    $L^2$-distance     | $\mathcal{F}(\nu^{est})$   |
> > |:----------|:--------:|:----------------------------:|:-----------------------:|:-----------------------------:|
> > |WDHA   |             |    $3.758\times 10^{-8}$   |  0.4869                     | $9.001 \times 10^{-2}$    |
> > |CWB     |   0.003 |    $2.98\times 10^{-4}$    | 1.321                        | $9.031 \times 10^{-2}$      |
> > | CWB    |  0.002  |    $2.538 \times 10^{-2}$  | 3.041                       |            0.1154                     |
> > |DSB      | 0.005   | $1.2 \times 10^{-4}$         | 0.8219                     | $9.013\times 10^{-2}$       |
> > |   DSB   | 0.004.  | $1.164\times 10^{-2}$.     | 2.281                       | 0.1016                                |
> >
> > We mark that the barycenter functional value of the truth is 0.09. This simulation is included in the appendix subsection C.1 of the revised version.
> >
> > *6.  How much critical is the compactness assumption on the input probability distributions for your method?*
> >
> > **Ans**: in practice, we can use finite grid points to approximate unbounded support, but this will incur some truncation losses.
> >
> > *7. Assumptions in Theorem 1 suggest that the input distributions should have the same support (although it is not verified in the very first experiment). Could you improve this assumption to consider a case of distinct supports?*
> >
> > **Ans**: in example 1, we put four shapes in the corners solely for visualization purpose, we can enlarge and center the shapes, the results would be the same. In addition, it's possible to separate the domains of the measures by following procedures in \cite{chartrand2009gradient} and considering the following formulation
> > $$
> > \mathcal{I}\_{\nu}^{\mu\_i}(\varphi\_i) = \int\_\Omega\frac{\|x\|\_2^2}{2} - \varphi\_i(x)d\nu + \int\_{\Phi\_i} \frac{\|x\|\_2^2}{2} - \varphi\_i^\ast d \mu\_i(x),
> > $$
> > where the support of $\nu$ is in $\Omega$, the suport of $\mu\_i$ is in $\Phi\_i$ and $\varphi\_i : \Omega \rightarrow \mathbb{R}$. The convex conjugate of $\varphi$ is defined to be $\varphi^{*}(y) = \sup\_{x \in \Omega} x^T y - \varphi(x)$ for any $y \in \Phi\_i$. It's then possible to extend the results by using Proposition 2.1 and Lemma 2.2 in Chartrand et al. [2009].
> >
> > *8.  Could you explain what prevents the term $\mathcal{F}_{\alpha, \beta}(\nu^{T+1})$ to be arbitrarily large in the upper bound of Theorem 1, Which may hurt the convergence rate?*
> >
> > **Ans**: this is true because all measures are assumed to be bounded above and have compact supports. Thus, the Wasserstein distance between $\nu^{T+1}$ and $\mu\_i$ is uniformly bounded for all $i$.

---

> > > ### Author Response · Authors · 2024-11-24
> > > **Response to Reviewer yaTH**
> > >
> > > *9. Could you detail how you efficiently implement the computation of the convex conjugate and the negative inverse Laplacian in the case of probability density functions discretized over a grid?*
> > >
> > > **Ans**: we note that for the 1D case, convex conjugate can be computed efficiently using the method in Corrias [1996]. For the 2D case, notice that
> > > $$
> > > \begin{aligned}
> > >     \varphi^{*}(y\_1, y\_2) :&= \sup\_{x\_1, x\_2} (x\_1-y\_1)^2/2 + (x\_2-y\_2)^2/2 - \varphi(x\_1, x\_2) \\
> > > \end{aligned}
> > > $$
> > >
> > > $$
> > >    = \sup\_{x\_1} \left( (x\_1-y\_1)^2/2 + \sup\_{x\_2} \\{ (x\_2-y\_2)^2/2 - \varphi(x\_1, x\_2) \\} \right) \\
> > > $$
> > >
> > > $$
> > >     = \sup\_{x\_1} \left( (x\_1-y\_1)^2/2 +  [\varphi(x\_1, \cdot )]^{*}(y\_2)  \right)
> > > $$
> > >
> > > Thus, the convex conjugate in the 2D case can be computed by iteratively applying the 1D solver to each row and column. Poisson's equation ($\Delta f = u$) with zero Neumann boundary conditions is classical and can be solved efficiently using fast Fourier transform, and we refer to chapter 3 of [Taylor, 1996] for detailed derivations of using Fourier transform to solve Poisson's equations. We have added more details on the computation of $\nabla \varphi$, $\varphi^{*}$ and $(\nabla \varphi)\_{\\#} \nu$ in the appendix subsection D.2 of the revised version
> > >
> > > *10. How do you compute the value function for competing methods?*
> > >
> > > **Ans**: once we have an estimated barycenter $\hat{\nu}$, we apply the back-and-forth approach Jacobs and L´eger
> > > [2020] to compute the Wasserstein distance between $\hat{\nu}$ and each $\mu\_i$.
> > >
> > >
> > > **References**
> > >
> > > - Robert J Berman. Convergence rates for discretized Monge–Amp`ere equations and quantitative stability of
> > > optimal transport. Foundations of Computational Mathematics, 21(4):1099–1140, 2021.
> > > - Rick Chartrand, Brendt Wohlberg, Kevin Vixie, and Erik Bollt. A gradient descent solution to the monge-
> > > kantorovich problem. Applied Mathematical Sciences, 3(22):1071–1080, 2009.
> > > - Lucilla Corrias. Fast legendre–fenchel transform and applications to hamilton–jacobi equations and conser-
> > > vation laws. SIAM journal on numerical analysis, 33(4):1534–1558, 1996.
> > > - Matt Jacobs and Flavien L´eger. A fast approach to optimal transport: The back-and-forth method. Nu-
> > > merische Mathematik, 146(3):513–544, 2020.
> > > - Matt Jacobs, Wonjun Lee, and Flavien L´eger. The back-and-forth method for wasserstein gradient flows.
> > > ESAIM: Control, Optimisation and Calculus of Variations, 27:28, 2021.
> > > - Alexander Korotin, Vage Egiazarian, Lingxiao Li, and Evgeny Burnaev. Wasserstein iterative networks for
> > > barycenter estimation. In Advances in Neural Information Processing Systems, volume 35, pages 15672–15686. Curran Associates, Inc., 2022.
> > > - Lingxiao Li, Aude Genevay, Mikhail Yurochkin, and Justin M Solomon. Continuous regularized wasserstein
> > > barycenters. Advances in Neural Information Processing Systems, 33:17755–17765, 2020.
> > > - Tianyi Lin, Chi Jin, and Michael Jordan. On gradient descent ascent for nonconvex-concave minimax
> > > problems. In International Conference on Machine Learning, pages 6083–6093, 2020.
> > > - Andrea Simonetto. Smooth strongly convex regression. In 2020 28th European Signal Processing Conference,
> > > pages 2130–2134, 2021.
> > > - Michael Eugene Taylor. Partial differential equations. 1, Basic theory. Springer, 1996.

---

> > > > ### Comment · Reviewer_yaTH · 2024-11-25
> > > > **Answer to the rebuttal**
> > > >
> > > > Thank you for the precise answers. I have some remarks regarding the response:
> > > >
> > > > - About the burden of dimension: I would suggest the authors to insist that their method is only applicable to **2D settings with compact support** (unless numerical evidence of the contrary), I don't think this is highlighted enough. Since the direction of the OT community tends to consider high-dimensional problems (with neural networks as mentioned by the authors), this has to be clearly stated either in the abstract/setting or as a limitation of the curent approach. In particular, it is not precised by the authors how the computation of the convex conjugate may be done in dimension $d>2$...
> > > >
> > > > - About the related work: I would suggest to also include [1] in the discussion.
> > > >
> > > > - About the uncertainty quantification: sorry if I was unclear, but displaying the average performance of the competing methods does not sufficiently reflect this uncertainty; the **standard deviations** should at least appear as it commonly expected in ML papers, see [2] for instance. This should be done each time a metric is displayed, ie, including the experiments of the main paper (for instance, the value functions are really close to each other in the first experiment) . Do you have these values ? Moreover, why did you not include the other methods for this 1D experiment ? Could you do more than 1d while still having access to the ground truth ?
> > > >
> > > > - About the additional experiment: thank you for this additional result, where a ground truth is available and which demonstrates the superiority of the current method. Nonetheless, I wonder why the authors chose this uncommon setting (where each input measure has the same structure, and therefore may be too 'toyish'), while most of papers consider a Gaussian setting, also proposed by Reviewer eGnH, which may bring more complexity, since the covariances may be different for each input measure, and may also enable to scale in dimension ?
> > > >
> > > > [1] Continuous Wasserstein-2 Barycenter Estimation without Minimax Optimization. Korotin et al. 2021
> > > >
> > > > [2]  Continuous Regularized Wasserstein Barycenters. Li et al. 2020.

---

> > > > > ### Author Response · Authors · 2024-11-25
> > > > > **Response to Reviewer yaTH**
> > > > >
> > > > > We want to thank reviewer yaTH for your consideration and quick responses.
> > > > >
> > > > > *(a) About the burden of dimension: I would suggest the authors to insist that their method is only applicable to 2D settings with compact support (unless numerical evidence of the contrary), I don't think this is highlighted enough. Since the direction of the OT community tends to consider high-dimensional problems (with neural networks as mentioned by the authors), this has to be clearly stated either in the abstract/setting or as a limitation of the curent approach.*
> > > > >
> > > > > **Ans**: thank you, our methods porvide more accurate computation for the low-dimensional case. The computation burden, to a large extent, depends on the total number of grid points.  (Jacobs \& L´eger,2020) used spatial grids as large as $4096\times 4096$ (2D) and $384 \times 384 \times 384$ (3D). We added right after stating the contributions on line 95 of the revised version: ``For limitations, the current approach is mainly limited to computing the Wasserstein barycenter of 2D or 3D distributions supported on a compact domain".
> > > > >
> > > > > *(b) In particular, it is not precised by the authors how the computation of the convex conjugate may be done in dimension $d > 2$.*
> > > > >
> > > > > **Ans**: we note that the procedure we described in response to point 9 is generalizable to arbitraty dimensions as long as we have a seperable cost function. We illustrate using the 3D case. The cost function for our case is seperable as $\|x-y\|\_2^2/2 = (x\_1-y\_1)^2/2 + (x\_2-y\_2)^2/2 + (x\_3-y\_3)^2/2$. Then,
> > > > > $$
> > > > > \varphi^{*}(y\_1, y\_2, y\_3):= \sup_{x\_1, x\_2} (x\_1-y\_1)^2/2 + (x\_2-y\_2)^2/2 +(x\_3-y\_3)^2/2 - \varphi(x\_1, x\_2, x\_3)
> > > > > $$
> > > > >
> > > > > $$
> > > > >   = \sup\_{x\_1, x\_2} \left( (x\_1-y\_1)^2/2 + (x\_2-y\_2)^2/2 + \sup\_{x\_3} \left\\{ (x\_3-y\_3)^2/2 - \varphi(x\_1, x\_2, x\_3) \right\\} \right)
> > > > > $$
> > > > >
> > > > > $$
> > > > > = \sup\_{x\_1} \left( (x\_1-y\_1)^2/2 + \sup\_{x\_2} \left\\{ (x\_2-y\_2)^2/2+ [\varphi(x\_1, x\_2, \cdot )]^{*}(y\_3) \right\\}  \right)
> > > > > $$
> > > > >
> > > > > $$
> > > > > = \sup\_{x\_1} \left( (x\_1-y\_1)^2/2 +  [ [ \varphi (x\_1, \cdot, \cdot ) ]^{*} ( y\_3 ) ]^{\ast}  (y\_2)  \right),
> > > > > $$
> > > > > which can be computed by applying the 1D solver to each dimension iteratively.
> > > > >
> > > > > *(c) About the related work: I would suggest to also include [1] in the discussion.*
> > > > >
> > > > > **Ans**: we apologize for missing this reference of scalable algorihtms, and have added ``To break the curse of dimensionality, scalable algorithms using input convex neural networks  \citep{korotin2021continuous} and  generative models \citep{NEURIPS2022_6489f2c6} have been investigated for the Wasserstein barycenter problem" on line 59 of the revised version.

---

> > > > > > ### Author Response · Authors · 2024-11-25
> > > > > > **Response to Reviewer yaTH**
> > > > > >
> > > > > > We want to thank reviewer yaTH for your consideration and quick responses.
> > > > > >
> > > > > > *(d) About the uncertainty quantification: sorry if I was unclear, but displaying the average performance of the competing methods does not sufficiently reflect this uncertainty; the standard deviations should at least appear as it commonly expected in ML papers, see [2] for instance. This should be done each time a metric is displayed, ie, including the experiments of the main paper (for instance, the value functions are really close to each other in the first experiment) . Do you have these values ? Moreover, why did you not include the other methods for this 1D experiment ? Could you do more than 1d while still having access to the ground truth?*
> > > > > >
> > > > > > **Ans**: Thank you. We added the standard deviations and included the results of CWB and DSB. We revised the writing in appendix subsection C.1 to ``Algorithm 3 (with $\alpha=10^{-3}, \beta=10^3$) performs slightly better with $\mathcal{W}_2$-distance $7.610\times 10^{-5}$ (standard deviation: $3.367\times 10^{-5}$) and $L^2$-distance 0.608 (standard deviation: $0.194$), while Algorithm 4 has $\mathcal{W}_2$-distance $7.771\times 10^{-5}$ (standard deviation: $3.32 \times 10^{-5}$) and $L^2$-distance 0.6434 (standard deviation: $0.451$). In addition, CWB has $\mathcal{W}_2$-distance $0.0022$ (standard deviation: $5.85 \times 10^{-4}$) and $L^2$-distance 0.082 (standard deviation: $0.039$). DSB achieves $\mathcal{W}_2$-distance $1.225 \times 10^{-5}$ (standard deviation: $2.644 \times 10^{-5}$) and $L^2$-distance 0.0699 (standard deviation: $0.034$). DSB performs the best in Wasserstein distance for this example". We didn't include CWB and DSB initially, because our primary goal is to demonstrate that Algorithms 3 and 4 perform similarly. For 1D distributions, the true barycenter can be easily computed by averaging the quantile functions, while the barycenter of 2D distributions with compact support are generally unknown, please also refer our response to point (e) below for the 2D Gaussian case.
> > > > > >
> > > > > > *(e) About the additional experiment: thank you for this additional result, where a ground truth is available and which demonstrates the superiority of the current method. Nonetheless, I wonder why the authors chose this uncommon setting (where each input measure has the same structure, and therefore may be too 'toyish'), while most of papers consider a Gaussian setting, also proposed by Reviewer eGnH, which may bring more complexity, since the covariances may be different for each input measure, and may also enable to scale in dimension ?*
> > > > > >
> > > > > > **Ans**: Our method works on distributions with compact support, because it needs to discretize the support and operates on the grid points. Technically, we first need to truncate Gaussian distributions to have compact support. The true barycenter of truncated Gaussian distributions are actually unknown.

---

> > > > > > > ### Comment · Reviewer_yaTH · 2024-11-26
> > > > > > > **Re: Answer to rebuttal**
> > > > > > >
> > > > > > > Thank you for pursuing the discussion and adding the missing details. I am willing to increase my score regarding the changes that have been made.

---

> > > > > > > > ### Author Response · Authors · 2024-11-26
> > > > > > > > **Response to Reviewer yaTH**
> > > > > > > >
> > > > > > > > We sincerely appreciate your constructive comments and time!

---

### Official Review · Reviewer_eGnH · 2024-11-03

**Soundness:** 2
**Presentation:** 3
**Contribution:** 2
**Rating:** 6
**Confidence:** 3

**Summary:**

This paper considers the problem of finding the unregularized barycenter between discrete probability measures. The authors reformulate this problem as a saddle-point optimization problem and propose a kind of gradient descent-ascent algorithm to solve it. The established algorithm is supported by theoretical investigation of its convergence rates. The approach is tested in two experimental setups where it is claimed to achieve better performance than two other baselines in terms of accuracy and time complexity.

**Strengths:**

The authors propose an interesting approach to solving the unregularized barycenter problem using the Wasserstein and $\dot{\mathbb{H}}^1$ gradients. The proposed approach outperforms the chosen competitors in time and space complexity.

**Weaknesses:**

My major concern is related to some discrepancy between the theoretical investigations of the proposed approach and its use in practical tasks. Namely, in Section 3.4, the authors prove several theoretical results regarding their method, including the important Theorem 1, which shows that the proposed algorithm allows finding stationary points of the objective functional. From the proofs in the Appendix, one can see that all theoretical results are logically connected, i.e. Lemma 2 uses the results of Lemma 1, Lemma 3 uses Lemma 2, and so on. Thus, the fulfillment of Theorem 1 also depends on the fulfillment of Lemma 1. At the same time, in Section 3.5 the authors propose to modify the algorithm for practical use in order to reduce the time complexity. This modification consists in replacing the projecting the potentials $\phi$ on $\mathbb{F}_{\alpha, \beta}$ with computing the second convex conjugate $(\phi)^{**}$. Such a replacement entails *non-fulfilment* of Lemma 1, as the authors themselves write in lines 422-423. Accordingly, Theorem 1 is also no longer true, i.e. the convergence of the algorithm's solutions to ground-truth solutions is no longer theoretically justified. I will appreciate if the authors could comment on the generalization of Theorem 1 to the case where Lemma 1 is not satisfied or conduct experiments with the original version of the algorithm.

The practical evaluation also lacks an experiment showing that the proposed algorithm (modified version) actually gives the **ground-truth solutions** for the considered barycenter problem. The authors only perform a comparison of some functional values ​​computed using a back-and-forth approach (Jacobs and Le ́geret, 2020) with the two baseline approaches $-$ WB (Solomon et al, 2015) and DSB (Janati et al., 2020). However, I have doubts regarding this metric for comparison, i.e., the computed functional values. It is not clear from the text how these values ​​are calculated - more details about the back-and-forth approach (Jacobs and Le ́geret, 2020) should be included. This is especially important since the approaches the authors compare with are entropy-regularized. Besides, I recommend the authors to conduct a comparison with ground truth solutions of the barycenter problem which can be constructed for the case of gaussian measures using, e.g., the iterative algorithm from (Álvarez-Esteban, Pedro C., et al., 2016), and perform the comparison with the baselines in this setup.

Moreover, I have some doubts regarding the significance of the developed algorithm since it borrows ideas which are already present in the field. $\dot{\mathbb{H}}^1$-gradient ascent algorithm for finding the maximizers of dual OT problem was proposed by (Jacobs & Le ́ger,2020), while the Wasserstein gradient for solving the barycenter problem was initially proposed in (Zemel & Panaretos, 2019; Chewi et al., 2020). The algorithm in the current work combines these two ideas.

**In summary**, I think that the contribution of the paper is questionable since the algorithm combines the ideas which are already present in the field and, thus, to represent interest for the community it should have some theoretically and practically justified advantages over the previous approaches. However, in the current version of the paper, it is not clear whether the modified algorithm which was used in the experimental part actually learns the ground-truth barycenter after the considered number of epochs or not. The theoretical investigations (convergence analysis) are not valid for this modified algorithm, while the experimental section also lacks the setup with known ground-truth solutions which can be used to mitigate this concern.

Minor:
 - line 132: descent $\rightarrow$ ascent;
- mistake in line 269 - there should be $\max$ instead of $\min$ over potentials.

**Questions:**

- The idea in lines 290-291 is not clear: “the definitions in subsection 2.3 imply that the Wasserstein gradient of J is given by $\nabla J (\nu, \phi) = id −\nabla\phi$, where …”. Please explain how this Wasserstein gradient is derived?

- Why do you consider only two baseline models, i.e., CWB (Solomon et al, 2015) and DSB (Janati et al., 2020) approaches? These models use the entropy regularization which makes the comparison not entirely fair (yet, I agree that it can be done in the case of small regularization parameter $\varepsilon$). Meanwhile, you have mentioned many other approaches which compute the barycenter in an unregularized setting, e.g., (Zemel & Panaretos, 2019; Chewi et al., 2020). Why do you not compare with these algorithms? I will highly appreciate it if you perform the comparison with some of these approaches during the rebuttal phase.
- What do you mean by $\nabla_1$ in line 211?
- In lines 315-318 you write: “we can alternatively use the L2 -gradient to update $\nu . However, in simulations, the Wasserstein gradient update significantly outperforms the L2-gradient update.” Can you provide an experiment justifying this claim?

**References.**

Álvarez-Esteban, P. C., Del Barrio, E., Cuesta-Albertos, J. A., & Matrán, C. (2016). A fixed-point approach to barycenters in Wasserstein space. Journal of Mathematical Analysis and Applications, 441(2), 744-762.

Matt Jacobs and Flavien Le ́ger. A fast approach to optimal transport: The back-and-forth method. Numerische Mathematik, 146(3):513–544, 2020.

Hicham Janati, Marco Cuturi, and Alexandre Gramfort. Debiased Sinkhorn barycenters. In Pro- ceedings of the 37th International Conference on Machine Learning, pp. 4692–4701, 2020.

Sinho Chewi, Tyler Maunu, Philippe Rigollet, and Austin J Stromme. Gradient descent algorithms for Bures-Wasserstein barycenters. In Conference on Learning Theory, pp. 1276–1304, 2020.

Yoav Zemel and Victor M. Panaretos. Fre ́chet means and Procrustes analysis in Wasserstein space. Bernoulli, 25(2):932 – 976, 2019.

Justin Solomon, Fernando de Goes, Gabriel Peyre ́, Marco Cuturi, Adrian Butscher, Andy Nguyen, Tao Du, and Leonidas Guibas. Convolutional Wasserstein distances: Efficient optimal transporta- tion on geometric domains. ACM Transactions on Graphs, 34(4), 2015.

---

> ### Author Response · Authors · 2024-11-24
> **Response to Reviewer eGnH**
>
> We sincerely appreciate your time, thoughtful consideration, and constructive guidance on our paper. Following your and the other referees' valuable comments, we added three additional simulation studies in the appendix of the revised version to address your questions. If we have successfully addressed some of your concerns, we would kindly ask you to consider reflecting this in your rating. Thank you again for your invaluable feedback.
>
> Below we provide point-by-point response to your comments in italics.
>
> *1,  My major concern is related to some discrepancy between the theoretical investigations of the proposed approach and its use in practical tasks. Namely, in Section 3.4, the authors prove several theoretical results regarding their method, including the important Theorem 1, which shows that the proposed algorithm allows finding stationary points of the objective functional. From the proofs in the Appendix, one can see that all theoretical results are logically connected, i.e. Lemma 2 uses the results of Lemma 1, Lemma 3 uses Lemma 2, and so on. Thus, the fulfillment of Theorem 1 also depends on the fulfillment of Lemma 1. At the same time, in Section 3.5 the authors propose to modify the algorithm for practical use in order to reduce the time complexity. This modification consists in replacing the projecting the potentials on with computing the second convex conjugate. Such a replacement entails non-fulfilment of Lemma 1, as the authors themselves write in lines 422-423. Accordingly, Theorem 1 is also no longer true, i.e. the convergence of the algorithm's solutions to ground-truth solutions is no longer theoretically justified. I will appreciate if the authors could comment on the generalization of Theorem 1 to the case where Lemma 1 is not satisfied or conduct experiments with the original version of the algorithm.*
>
> **Ans**: Thank you for your comments. The same discrepancy issue also exists in other works that utilize the $\mathbb{H}^1$-gradient if any convergence theory is to be developed; see the assumptions in Proposition 1 [Jacobs and Leger, 2020] and Theorem 2.8 [Jacobs et al., 2021. We believe that the satisfying empirical performance of our algorithm should not be discounted merely because of this. Please check our additional simulation result with ground truth. Without additional regularities, the generalization of the Theorem can be challenging. The main obstacle is that the map $\nu \rightarrow T_{\nu}^{\mu}$ is not Lipschitz; see [Berman, 2021] for more details regarding this.
>
> In practice, the projection can be computed following the approach developed by Simonetto [2021], which is implemented as the Python function `reg4opt.regression.convex\_regression` in library `reg4opt`. Unfortunately, this projection algorithm requires solving a large  convex quadratically constrained quadratic problem so that it takes weeks to run an experiment in 2D.
>
> Instead, we implement the algorithm with projection (Algorithm 3) and test the method on 1D distributions. More precisely, for each repetition $t$ and $i=1,2,3$, we let $\mu\_{i,t}$ be the truncated distribution of $N(a\_i,{\sigma\_{i}}^2)$ on the domain $[0,1]$, where $a\_i \sim \text{uniform}\left[0.3,0.7\right]$ and  $\sigma\_i \sim \text{uniform}\left[0,1\right]$. Let $\overline{\nu}\_t$, $\overline{\nu}\_{1, t}$, $\overline{\nu}\_{2, t}$
> be the true barycenter, the barycernter computed from Algorithm 3 (with projection onto $\mathbb{F}\_{\alpha, \beta}$), and the barycernter computed from Algorithm 4 (with double convex conjugates) respectively. We repeat the experiment $T=300$ times and report two types of average distances between the groundtruth and each computed barycenters : average $\mathcal{W}\_2$-distance $\frac{1}{T} \sum\_{t=1}^T \mathcal{W}_2(\overline{\nu}\_t, \overline{\nu}\_{j, t})$ and average $L^2$-distance between densities $ \frac{1}{T} \sum\_{t=1}^T (\int (\overline{\nu}\_t(x)- \overline{\nu}\_{j, t}(x))^2 \; d x)^{1/2}, j=1,2$. Algorithm 3 performs slightly better with $\mathcal{W}_2$-distance $7.610\times 10^{-5}$ and $L^2$-distance 0.608, while Algorithm 4 has $\mathcal{W}\_2$-distance $7.771\times 10^{-5}$ and $L^2$-distance 0.6434. We have included this study in subsection C.2 in the appendix of the revised version.

---

> > ### Author Response · Authors · 2024-11-24
> > **Response to Reviewer eGnH**
> >
> > *2. The practical evaluation also lacks an experiment showing that the proposed algorithm (modified version) actually gives the ground-truth solutions for the considered barycenter problem. The authors only perform a comparison of some functional values-computed using a back-and-forth approach (Jacobs and L\'{e}geret, 2020) with the two baseline approaches WB (Solomon et al, 2015) and DSB (Janati et al., 2020). However, I have doubts regarding this metric for comparison, i.e., the computed functional values. It is not clear from the text how these values-are calculated - more details about the back-and-forth approach (Jacobs and L\'{e}geret, 2020) should be included. This is especially important since the approaches the authors compare with are entropy-regularized. Besides, I recommend the authors to conduct a comparison with ground truth solutions of the barycenter problem which can be constructed for the case of gaussian measures using, e.g., the iterative algorithm from (\'{A}lvarez-Esteban, Pedro C., et al., 2016), and perform the comparison with the baselines in this setup.*
> >
> > **Ans**: that's a good point. We added a new simulation in the revised version for which the ground truth is known. This simulation demonstrates that both CWB and DSB may output blurry barycenters, while our result is visually no different from the truth. We, in addition, report the $L^2$-distance between computed barycenter densities and the true density.
> >
> > In this new simulation, the goal is to compute the barycenter of four uniform distributions supported on round disks of radius 0.15, centered at $(0.2, 0.2)$, $(0.2, 0.8)$, $(0.8, 0.2)$, $(0.8, 0.8)$ respectively. It's clear that the true barycenter is uniform on the disk of radius 0.15 centered at $(0.5, 0.5)$. The computed barycenter densities by WDHA, CWB and DSB are shown in Figure 3 of the revised version. We note that regularization parameters reg= 0.003 and reg=0.005 are the optimal choices for CWB and DSB respectively. Smaller regularization parameters lead nonconvergent and worse results for both CWB and DSB. We report in the following table the Wasserstein distance between computed barycenter distribution to the truth, $L^2$-distance between computed barycenter densities and the true density, and the barycenter functional value. Our method is uniformly the best, and in particular, the improvement in the Wasserstein distance is of orders of magnitude.
> >
> > |               |   reg     | Wasserstein distance      |    $L^2$-distance     | $\mathcal{F}(\nu^{est})$   |
> > |:----------|:--------:|:----------------------------:|:-----------------------:|:-----------------------------:|
> > |WDHA   |             |    $3.758\times 10^{-8}$   |  0.4869                     | $9.001 \times 10^{-2}$    |
> > |CWB     |   0.003 |    $2.98\times 10^{-4}$    | 1.321                        | $9.031 \times 10^{-2}$      |
> > | CWB    |  0.002  |    $2.538 \times 10^{-2}$  | 3.041                       |            0.1154                     |
> > |DSB      | 0.005   | $1.2 \times 10^{-4}$         | 0.8219                     | $9.013\times 10^{-2}$       |
> > |   DSB   | 0.004.  | $1.164\times 10^{-2}$.     | 2.281                       | 0.1016                                |
> >
> > We mark that the barycenter functional value of the truth is 0.09. This simulation is included in the appendix subsection C.1 of the revised version.
> >
> > Thank you for pointing out  (\'{A}lvarez-Esteban, Pedro C., et al., 2016). We have added the following on line 47 of the revised version ``Alvarez-Esteban et al. (2016) propose a fixed point approach that is effective for any location-scatter family."
> >
> > *3. Moreover, I have some doubts regarding the significance of the developed algorithm since it borrows ideas which are already present in the field. $\mathbb{H}^1$-gradient ascent algorithm for finding the maximizers of dual OT problem was proposed by (Jacobs \& L\'{e}ger,2020), while the Wasserstein gradient for solving the barycenter problem was initially proposed in (Zemel \& Panaretos, 2019; Chewi et al., 2020). The algorithm in the current work combines these two ideas.*
> >
> > **Ans**: We proposed a gradient-ascent type algorithm for solving the nonconvex-concave minimax formulation of the Wasserstein barycenter problem. Our method works beyond Gaussian distributions. The work of (Chewi et al., 2020) focus on Gaussian distributions, as the optimal transport maps have closed forms in this case. A work that can be considered as combination of the two ideas is to solve the inner maximization problem completely by applying $\mathbb{\dot{H}}^1$-gradient descent until it converges, while our method updates the current potential with just one round of $\mathbb{\dot{H}}^1$-gradient descent.

---

> > > ### Author Response · Authors · 2024-11-24
> > > **Response to Reviewer eGnH**
> > >
> > > *4. The idea in lines 290-291 is not clear: “the definitions in subsection 2.3 imply that the Wasserstein gradient of $J$ is given by $ \nabla J(\nu, \phi) = id - \nabla \phi $, where ...". Please explain how this Wasserstein gradient is derived?*
> > >
> > > **Ans**:To compute Wasserstein gradient of $\mathcal{J}(\nu, \varphi)$, we apply the definition in Subsection 2.3. Note that
> > >
> > > $$
> > > \begin{aligned}
> > >  &  \left. \frac{d}{d \varepsilon} \mathcal{J}(\nu + \varepsilon \chi, \phi) \right|_{\varepsilon =0}    \\
> > >     =&   \left. \frac{d}{d \varepsilon} \left( \frac{1}{n} \sum\_{i=1}^n \int\_\Omega\frac{\|x\|\_2^2}{2} - \varphi_i(x) d (\nu +\varepsilon \chi ) + \int\_\Omega\frac{\|x\|\_2^2}{2} - \varphi\_i^\ast  \mu\_i(x) \right) \right|\_{\varepsilon =0}
> > > \end{aligned}
> > > $$
> > >
> > > $$
> > > \begin{aligned}
> > >  &  \left. \frac{d}{d \varepsilon} \mathcal{J}(\nu + \varepsilon \chi, \phi) \right|_{\varepsilon =0}    \\
> > >    = &  \left. \frac{d}{d \varepsilon} \left( \varepsilon \frac{1}{n} \sum\_{i=1}^n \int\_\Omega\frac{\|x\|\_2^2}{2} - \varphi\_i(x) d  \chi   \right) \right|\_{\varepsilon =0}
> > > \end{aligned}
> > > $$
> > >
> > > $$
> > > \begin{aligned}
> > >  &  \left. \frac{d}{d \varepsilon} \mathcal{J}(\nu + \varepsilon \chi, \phi) \right|_{\varepsilon =0}    \\
> > >  = &  \left. \frac{d}{d \varepsilon} \left( \varepsilon  \int\_\Omega\frac{\|x\|\_2^2}{2} - \overline{\varphi}(x) d \chi   \right) \right|\_{\varepsilon =0}  \\
> > >   = & \int\_\Omega\frac{\|x\|\_2^2}{2} - \overline{\varphi}(x) d  \chi ,
> > > \end{aligned}
> > > $$
> > >
> > > which implies that $\frac{\delta \mathcal{J}}{\delta\mu}(\mu)(x) = \frac{\|x\|\_2^2}{2} - \overline{\varphi}(x)$. By the definition of Wasserstein gradient, Wasserstein gradient of $\mathcal{J} = \nabla \frac{\delta \mathcal{J}}{\delta\mu}(\mu) = id - \nabla \overline{\varphi}$. We have added this part to the appendix subsection D.1 of the revised version.
> > >
> > >
> > > *5.  Why do you consider only two baseline models, i.e., CWB (Solomon et al, 2015) and DSB (Janati et al., 2020) approaches? These models use the entropy regularization which makes the comparison not entirely fair (yet, I agree that it can be done in the case of small regularization parameter $\varepsilon$). Meanwhile, you have mentioned many other approaches which compute the barycenter in an unregularized setting, e.g., (Zemel \& Panaretos, 2019; Chewi et al., 2020). Why do you not compare with these algorithms? I will highly appreciate it if you perform the comparison with some of these approaches during the rebuttal phase.*
> > >
> > > **Ans**: We note that the algorithms in (Zemel \& Panaretos, 2019; Chewi et al., 2020) are not applicable to the examples in our paper, as they implicitly assume that in each iteration, the optimal transport between $\nu^{t}$ and each $\mu\_i$ are known, which is only true for Guassian distributions. We fill this gap by proposing the the \emph{Wasserstein-Descent $\dot{\mathbb{H}}^1$-Ascent} (WDHA) algorithm.
> > >
> > >
> > > *6. What do you mean by $\nabla\_{1}$ in line 211?*
> > >
> > > **Ans**: We have changed $\nabla\_{1}$ to $\nabla_{x}$ indicating gradient w.r.t $x$ in the revised version.

---

> > > > ### Author Response · Authors · 2024-11-24
> > > > **Response to Reviewer eGnH**
> > > >
> > > > *7. In lines 315-318 you write: ``we can alternatively use the L2 -gradient to update $\nu$ . However, in simulations, the Wasserstein gradient update significantly outperforms the L2-gradient update." Can you provide an experiment justifying this claim?*
> > > >
> > > > **Ans**: Certainly, we implemented this algorithm and tested it on uniform distributions supported on round disks for which the ground truth is known. Let $\nu(x)$, $\mu_i(x)$ be density functions. We can write
> > > > $$
> > > > \mathcal{J}(\nu, \phi) = \frac{1}{n} \sum\_{i=1}^n \int\_\Omega \left(\frac{\|x\|\_2^2}{2} - \varphi\_i(x) \right) \nu(x)  d x + \int\_\Omega \left( \frac{\|x\|\_2^2}{2} - \varphi\_i^\ast \right) \mu\_i(x) d x.
> > > > $$
> > > > Let $L^2(\lambda)=\\{ h : \int h(x)^2 d \lambda < \infty \\}$ be the space of functions that are $L^2$-integrable with respect to the Lebesgue measure $\lambda$ and $\mathcal{D} = \{ f \in L^2(\lambda): f(x) \geq 0, \int f(x) d \lambda =1 \}$ be the set of density functions. Notice that the $L^2$-gradient of $\mathcal{J}$ with respect to $\nu(x)$ is $\nabla\_{\nu} \mathcal{J}(\nu, \phi):= \frac{1}{n} \sum\_{i=1}^n \frac{\|x\|\_2^2}{2} - \varphi\_i(x)$, we may consider the $L^2$-descent $\mathbb{\dot{H}}^1$-ascent algorithm: for each iteration $t$, do
> > > >
> > > > - For $i=1,2, \dots, n$, compute
> > > > $$
> > > >     \begin{aligned}
> > > >     & \widehat{\varphi}\_{i}^{t+1} = \varphi\_{i}^{t} + \eta^t\_i \nabla\_{\varphi_i} \mathcal{J} (\nu^t, \varphi^{t}) \\
> > > >     & \varphi\_{i}^{t+1} = (\widehat{\varphi}\_{i}^{t+1})^{**}
> > > > \end{aligned}
> > > > $$
> > > > - Compute
> > > > $$
> > > > \begin{aligned}
> > > > \tilde{\nu}^{t+1} & =  \nu^{t} - \tau\_t \nabla \_{\nu} \mathcal{J}(\nu, \phi);
> > > >  \\
> > > > \nu^{t+1} & = \mathcal{P}_{\mathcal{D}}(\tilde{\nu}^{t+1})
> > > > \end{aligned}
> > > > $$
> > > >
> > > > Here, $ \mathcal{P}_{\mathcal{D}}(\tilde{\nu}^{t+1})$ is the projection of $\tilde{\nu}^{t+1}$ onto $\mathcal{D}$, which can be computed using python function `pyproximal.Simplex` in library `1PyProximal`. We apply this algorithm to the uniform distributions supported on round disks, and observe that it actually diverges leading to a wrong result. Nevertheless, we plot the output in Figure 4 of the revised version. This simulation is included in appendix subsection C.3 of the revised version.
> > > >
> > > > **References**
> > > > - Robert J Berman. Convergence rates for discretized Monge–Amp`ere equations and quantitative stability of
> > > > optimal transport. Foundations of Computational Mathematics, 21(4):1099–1140, 2021.
> > > > - Rick Chartrand, Brendt Wohlberg, Kevin Vixie, and Erik Bollt. A gradient descent solution to the monge-
> > > > kantorovich problem. Applied Mathematical Sciences, 3(22):1071–1080, 2009.
> > > > - Lucilla Corrias. Fast legendre–fenchel transform and applications to hamilton–jacobi equations and conser-
> > > > vation laws. SIAM journal on numerical analysis, 33(4):1534–1558, 1996.
> > > > - Matt Jacobs and Flavien L´eger. A fast approach to optimal transport: The back-and-forth method. Nu-
> > > > merische Mathematik, 146(3):513–544, 2020.
> > > > - Matt Jacobs, Wonjun Lee, and Flavien L´eger. The back-and-forth method for wasserstein gradient flows.
> > > > ESAIM: Control, Optimisation and Calculus of Variations, 27:28, 2021.
> > > > - Alexander Korotin, Vage Egiazarian, Lingxiao Li, and Evgeny Burnaev. Wasserstein iterative networks for
> > > > barycenter estimation. In Advances in Neural Information Processing Systems, volume 35, pages 15672–15686. Curran Associates, Inc., 2022.
> > > > - Lingxiao Li, Aude Genevay, Mikhail Yurochkin, and Justin M Solomon. Continuous regularized wasserstein
> > > > barycenters. Advances in Neural Information Processing Systems, 33:17755–17765, 2020.
> > > > - Tianyi Lin, Chi Jin, and Michael Jordan. On gradient descent ascent for nonconvex-concave minimax
> > > > problems. In International Conference on Machine Learning, pages 6083–6093, 2020.
> > > > - Andrea Simonetto. Smooth strongly convex regression. In 2020 28th European Signal Processing Conference,
> > > > pages 2130–2134, 2021.
> > > > - Michael Eugene Taylor. Partial differential equations. 1, Basic theory. Springer, 1996.

---

> ### Comment · Reviewer_eGnH · 2024-11-25
> **Response to Authors**
>
> Thank you for your detailed answers and new experiments. The newly conducted experiments indeed clarify some of my concerns. However, I agree with the reviewer yaTH that the fact that your approach is tested only in the experiments with dimension $d=2$, raises questions about its applicability to high dimensions.
>
> Still, I appreciate that the authors conduct the experiments which I suggested and increase my score to 6.

---

> > ### Author Response · Authors · 2024-11-26
> > **Response to Reviewer eGnH**
> >
> > We sincerely appreciate your constructive comments and time!

---

### Official Review · Reviewer_5oHW · 2024-11-05

**Soundness:** 3
**Presentation:** 3
**Contribution:** 3
**Rating:** 8
**Confidence:** 3

**Summary:**

In this paper, the authors present a new, computationally efficient approach for the Wasserstein barycenter problem. Without resorting to entropic-regularization, which degrades the quality of the result, they propose to formulate the problem as a nonconvex-concave minimax optimization problem, solved with a mixed gradient descent in the Wasserstein space and gradient ascent in Sobolev space. The algorithm is shown to converge to a fixed point with the classic rate for this class of problems. The final complexity is in $O(m \log m)$ if one projection step is approximated by a double conjugate (which violates the theory but performs well in practice), to be compared with $O(m^3)$ in most previous approaches. Experiments on synthetic data and handwritten digits are performed.

**Strengths:**

I liked this paper. The idea is natural but novel and efficient, and the presentation is clear and well-written despite the technicity of the work. Experiments are convincing. The authors are also honest in giving credit to inspirations from previous works (esp. Jacobs and Léger, and Lin et al.).

**Weaknesses:**

One point that would deserve more discussion is the property of the nonconvex-concave problem. Unless I am mistaken (I could very well miss something), by reformulating the problem as a nonconvex problem, one loses convergence to the true solution, but only towards a stationary point. This is a bit obfuscated in the current formulation, and would deserve a more honest discussion: efficient computation is obtained by "sacrificing" some theoretical guarantees, even with good practical performance. (If I missed something and convergence to the true barycenter is guaranteed, then it also deserved to be written!)

Minor typo: "dicuss", "Sythentic"

**Questions:**

See above.

---

> ### Author Response · Authors · 2024-11-24
> **Response to Reviewer 5oHW**
>
> We sincerely appreciate your time, thoughtful consideration, and constructive guidance on our paper. Following your and the other referees' valuable comments, we added three additional simulation studies in the appendix of the revised version to address your questions. If we have successfully addressed some of your concerns, we would kindly ask you to consider reflecting this in your rating. Thank you again for your invaluable feedback.
>
> Below we provide point-by-point response to your comments in italics.
>
> *1, One point that would deserve more discussion is the property of the nonconvex-concave problem. Unless I am mistaken (I could very well miss something), by reformulating the problem as a nonconvex problem, one loses convergence to the true solution, but only towards a stationary point. This is a bit obfuscated in the current formulation, and would deserve a more honest discussion: efficient computation is obtained by ``sacrificing" some theoretical guarantees, even with good practical performance. (If I missed something and convergence to the true barycenter is guaranteed, then it also deserved to be written!)*
>
> **Ans**: Thank you. In Euclidean space, the goal of minimax problem is to find the minimizer of $\Phi(x)$, where $\Phi(x) = \max_{y \in \mathcal{Y}} f(x, y)$. If $f(x, y)$ is nonconvex in $x$, then $\Phi(x)$ is nonconvex, and thus a feasible goal is to find $x$ such that $\| \nabla \Phi(x) \|_2$ is small.
>
> For the Wasserstein barycenter problem, by definition,  $\overline{\nu}$ is a Wasserstein barycenter if $id - \frac{1}{n} \sum_{i=1}^n \nabla \varphi_{\overline{\nu}}^{\mu_i}=0 $, where $\varphi_{\overline{\nu}}^{\mu_i}$ is the Kantorovich potential between $\overline{\nu}$ and $\mu_i$. If $\varphi_{\overline{\nu}}^{\mu_i} \in \mathbb{F}\_{\alpha, \beta}$ for all $i$, then $\overline{\nu}$ is a stationary point of $\mathcal{F}\_{\alpha, \beta}$, i.e.,  Wasserstein gradient of $\mathcal{F}\_{\alpha, \beta}(\overline{\nu})$ is 0. Reversely, if we assume that the Kantorovich potential between the true barycenter and each $\mu_i$ is in $\mathbb{F}\_{\alpha, \beta}$, then
> Wasserstein gradient of $\mathcal{F}\_{\alpha, \beta}(\overline{\nu}')$ is 0 would mean that $\overline{\nu}'$ is a Wasserstein barycenter. We have included this discussion in Remark 1 of the revised version.

---

> > ### Comment · Reviewer_5oHW · 2024-11-27
> > **Answer to rebuttal**
> >
> > Thank you for answering my questions. I keep my positive score.

---

> > > ### Author Response · Authors · 2024-11-27
> > > **Response to Reviewer 5oHW**
> > >
> > > We sincerely appreciate your constructive comments and time!

---

### Meta-Review · Area_Chair_ZRZH · 2024-12-17

**Metareview:**

This paper presents the Wasserstein-Descent–Ascent (WDHA) algorithm, a novel approach for computing unregularized Wasserstein barycenters via a nonconvex-concave minimax optimization framework. The key contribution lies in its primal-dual formulation that alternates between Wasserstein and Sobolev geometries, achieving computational efficiency for low-dimensional settings. While the method shows potential, several critical weaknesses lead to the recommendation of rejection. The numerical results are underwhelming, with artifacts indicating potential instability. The experiments are restricted to low-dimensional settings, raising concerns about scalability to higher-dimensional problems, which are central to optimal transport applications. Furthermore, the theoretical contributions are limited, with some theorems criticized as lacking explanatory power or practical relevance. Reviewers also noted insufficient acknowledgment of the method's limitations, such as its reliance on approximations that weaken theoretical guarantees. While the authors provided additional clarifications during the rebuttal, the core issues remain unresolved, making the work feel premature for acceptance at this stage. Addressing these gaps is crucial to solidify the contribution and ensure its robustness and applicability.

**Additional Comments On Reviewer Discussion:**

During the rebuttal period, reviewers raised concerns about numerical instability, limited scalability to high-dimensional settings, and the superficial nature of the theoretical contributions. The authors responded with additional simulations and clarifications, addressing some questions about their algorithm's behavior. However, critical issues, such as the robustness of the numerical results, scalability, and the practical relevance of the theoretical insights, were not sufficiently resolved. While the new experiments added some value, they did not mitigate the overarching concerns. After considering the discussion, I weighed the lack of stability and scalability as decisive factors in my recommendation for rejection, as these are fundamental for advancing research in this area.

---

### Decision · Program_Chairs · 2025-01-22

Reject